# SIMPLICITY BIAS IN 1-HIDDEN LAYER NEURAL NETWORKS

## ABSTRACT

Recent works Shah et al. (2020); Chen et al. (2021) have demonstrated that neural networks exhibit extreme *simplicity bias* (SB). That is, they learn *only the simplest* features to solve a task at hand, even in the presence of other, more robust but more complex features. Due to the lack of a general and rigorous definition of *features*, these works showcase SB on *semi-synthetic* datasets such as Color-MNIST, MNIST-CIFAR where defining features is relatively easier.

In this work, we rigorously define as well as thoroughly establish SB for *one hidden layer* neural networks. More concretely, (i) we define SB as the network essentially being a function of a low dimensional projection of the inputs (ii) theoretically, we show that when the data is linearly separable, the network primarily depends on only the linearly separable (1-dimensional) subspace even in the presence of an arbitrarily large number of other, more complex features which could have led to a significantly more robust classifier, (iii) empirically, we show that models trained on *real* datasets such as Imagenette and Waterbirds-Landbirds indeed depend on a low dimensional projection of the inputs, thereby demonstrating SB on these datasets, iv) finally, we present a natural ensemble approach that encourages diversity in models by training successive models on features not used by earlier models, and demonstrate that it yields models that are significantly more robust to Gaussian noise.

## 1 INTRODUCTION

It is well known that neural networks (NNs) are vulnerable to distribution shifts as well as to adversarial examples (Szegedy et al., 2014; Hendrycks et al., 2021). A recent line of work (Geirhos et al., 2018; Shah et al., 2020; Geirhos et al., 2020) proposes that *Simplicity Bias (SB)* – aka shortcut learning – i.e., the tendency of neural networks (NNs) to learn only the simplest features over other useful but more complex features, is a key reason behind this non-robustness. The argument is roughly as follows: for example, in the classification of swans vs bears, as illustrated in Figure 1, there

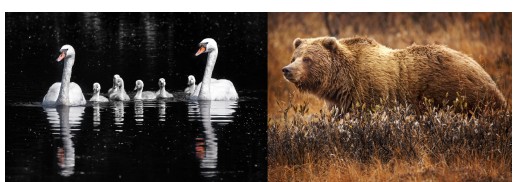

Figure 1: Classification of swans vs bears. There are several features such as background, color of the animal, shape of the animal etc., each of which is sufficient for classification but using all of them will lead to a more robust model. [1]

are many features such as background, color of the animal, shape of the animal etc. that can be used for classification. However using only one or few of them can lead to models that are not robust to specific distribution shifts, while using all the features can lead to more robust models.

Several recent works have demonstrated SB on a variety of *semi-real constructed datasets* (Geirhos et al., 2018; Shah et al., 2020; Chen et al., 2021), and have hypothesized SB to be the key reason for NN's brittleness to distribution shifts (Shah et al., 2020). However, such observations are still only for specific semi-real datasets, and a general method that can identify SB on a *given dataset* and a *given model* is still missing in literature. Such a method would be useful not only to estimate the robustness of a model but could also help in designing more robust models.

---

[1] Image source: Wikipedia swa, bea.

A key challenge in designing such a general method to identify (and potentially fix) SB is that the notion of *feature* itself is vague and lacks a rigorous definition. Existing works like Geirhos et al. (2018); Shah et al. (2020); Chen et al. (2021) avoid this challenge of vague feature definition by using carefully designed datasets (e.g., concatenation of MNIST images and CIFAR images), where certain high level features (e.g., MNIST features and CIFAR features, shape and texture features) are already baked in the dataset definition, and arguing about their *simplicity* is intuitively easy.

**Contributions**: One of the main contributions of this work is to provide a precise definition of a particular simplicity bias – LD-SB – of $1$-*hidden layer neural networks*. In particular, we characterize SB as *low dimensional input dependence* of the model. Concretely,

**Definition 1.1** (LD-SB). *A model $f : \mathbb{R}^d \to \mathbb{R}^c$ with inputs $x \in \mathbb{R}^d$ and outputs $f(x) \in \mathbb{R}^c$ (e.g., logits for $c$ classes), trained on a distribution $(x, y) \sim \mathcal{D}$ satisfies LD-SB if there exists a* projection *matrix $P \in \mathbb{R}^{d \times d}$ satisfying the following:*

- *rank $(P) = k \ll d$,*
- *$f(Px_1 + P_\perp x_2) \approx f(x_1) \ \forall (x_1, y_1), (x_2, y_2) \sim \mathcal{D}$, and*
- *An independent model $g$ trained on $(P_\perp x, y)$ where $(x, y) \sim \mathcal{D}$ achieves high accuracy.*

*Here $P^\perp$ is the projection matrix onto the subspace orthogonal to $P$.*

In words, LD-SB says that there exists a small $k$-dimensional subspace (given by the projection matrix $P$) in the input space $\mathbb{R}^d$, which is the only thing that the model $f$ considers in labeling any input point $x$. In particular, if we *mix* two data points $x_1$ and $x_2$ by using the projection of $x_1$ onto $P$ and the projection of $x_2$ onto the orthogonal subspace $P_\perp$, the output of $f$ on this *mixed point* $Px_1 + P_\perp x_2$ is the same as that on $x_1$. This would have been fine if the subspace $P_\perp$ does not contain any feature useful for classification. However, the third bullet point says that $P_\perp$ indeed contains features that are useful for classification since an independent model $g$ trained on $(P_\perp x, y)$ achieves high accuracy.

Furthermore, theoretically, we demonstrate LD-SB of $1$-hidden layer NNs for a fairly general class of distributions called *independent features model (IFM)*, where the features (i.e., coordinates) are distributed independently conditioned on the label. IFM has a long history and is widely studied, especially in the context of naive-Bayes classifiers Lewis (1998). For IFM, we show that as long as there is even a *single* feature in which the data is linearly separable, NNs trained using SGD will learn models that rely almost exclusively on this linearly separable feature, even when there are an *arbitrarily large number* of features in which the data is separable but with a *non-linear* boundary. Empirically, we demonstrate LD-SB on three real world datasets: binary and multiclass version of Imagenette (FastAI, 2021) as well as waterbirds-landbirds (Sagawa et al., 2020a) dataset. Compared to the results in Shah et al. (2020), our results (i) theoretically show LD-SB in a fairly general setting and (ii) empirically show LD-SB on real datasets.

Finally, building upon these insights, we propose a simple ensemble method – *OrthoP* – that sequentially constructs NNs by projecting out principle input data directions that are used by previous NNs. We demonstrate that this method can lead to significantly more robust ensembles for real-world datasets in presence of simple distribution shifts like Gaussian noise.

**Why only $1$-hidden layer networks?**: One might wonder why the results in this paper are restricted to $1$-hidden layer networks and why they are interesting. We present two reasons.

1. From a **theoretical** standpoint, prior works have thoroughly characterized the training dynamics of infinite width $1$-hidden layer networks under different initialization schemes (Chizat et al., 2019) and have also identified the limit points of gradient descent for such networks (Chizat & Bach, 2020). Our results crucially build upon these prior works. On the other hand, we do not have such a clear understanding of the dynamics of deeper networks.

2. From a **practical** standpoint, the dominant paradigm in machine learning right now is to pretrain large models on large amounts of data and then finetune on small target datasets. Given the large and diverse pretraining data seen by these models, it has been observed that they do learn rich features (Rosenfeld et al., 2022; Nasery et al., 2022). However, finetuning on target datasets might not utilize all the features in the pretrained model. Consequently, approaches that can train robust finetuning heads (such as a $1$-hidden layer network on top) can be quite effective.

Extending our results to deeper networks and to other architectures is an exciting direction of research from both theoretical and practical points of view.

**Paper organization**: This paper is organized as follows. Section 2 presents related work. Section 3 presents preliminaries. Our main results on LD-SB are presented in Section 4. Section 5 presents results on training diverse classifiers. We conclude in Section 6.

## 2 RELATED WORK

**Simplicity Bias**: Subsequent to Shah et al. (2020), there have been several papers investigating the presence/absence of SB in various networks as well as reasons behind SB Scimeca et al. (2021). Of these, Huh et al. (2021) and Galanti & Poggio (2022) are the most closely related works to ours. Huh et al. (2021) empirically observe that on certain synthetic datasets, the *embeddings* of NNs both at initialization as well as after training have a low rank structure. Galanti & Poggio (2022) provide a theoretical intuition behind the relation between various hyperparameters (such as learning rate, batch size etc.) and rank of learnt weight matrices, and demonstrate it empirically. In contrast, we prove LD-SB theoretically on the IFM model as well as empirically validate this on real datasets. Moreover, we also show how to use LD-SB to train a second diverse model and combine it to obtain a robust ensemble. Pezeshki et al. (2021) proposes that *gradient starvation* at the beginning of training is a potential reason for SB in the lazy/NTK regime but the conditions are hard to interpret. In contrast, our results are shown for any dataset in the IFM model in the *rich* regime of training. Finally Lyu et al. (2021) consider anti-symmetric datasets and show that single hidden layer input homogeneous networks (i.e., without *bias* parameters) converge to linear classifiers. However, such networks have strictly weaker expressive power compared to those with bias parameters.

**Learning diverse classifiers**: There have been several works that attempt to learn diverse classifiers. Most works try to learn such models by ensuring that the input gradients of these models do not align (Ross & Doshi-Velez, 2018; Teney et al., 2022). Xu et al. (2022) proposes a way to learn diverse/orthogonal classifiers under the assumption that a complete classifier, that uses all features is available, and demonstrates its utility for various downstream tasks such as style transfer. Lee et al. (2022) learns diverse classifiers by enforcing diversity on unlabeled target data.

**Spurious correlations**: There has been a large body of work which identifies the reasons for spurious correlations in NNs (Sagawa et al., 2020b) as well as proposing algorithmic fixes in different settings (Liu et al., 2021; Chen et al., 2020).

**Implicit bias of gradient descent**: There is also a large body of work understanding the implicit bias of gradient descent dynamics. Most of these works are for standard linear (Ji & Telgarsky, 2019) or deep linear networks (Soudry et al., 2018; Gunasekar et al., 2018). For nonlinear neural networks, one of the well-known results is for the case of 1-hidden layer neural networks with homogeneous activation functions (Chizat & Bach, 2020), which we crucially use in our proofs.

## 3 PRELIMINARIES

In this section, we provide the notation and background on infinite width max-margin classifiers that is required to interpret the results of this paper.

### 3.1 BASIC NOTIONS

**1-hidden layer neural networks and loss function.** Consider instances $x \in \mathcal{R}^d$ and labels $y \in \{\pm 1\}$ jointly distributed as $\mathcal{D}$. A 1-hidden layer neural network model for predicting the label for a given instance $x$, is defined by parameters $(\bar{w} \in \mathbb{R}^{m \times d}, \bar{b} \in \mathbb{R}^m, \bar{a} \in \mathbb{R}^m)$. For a fixed activation function $\phi$, given input instance $x$, the model is given as $f((\bar{w}, \bar{b}, \bar{a}), x) := \langle \bar{a}, \phi(\bar{w}x + \bar{b}) \rangle$, where $\phi(\cdot)$ is applied elementwise. The cross entropy loss $\mathcal{L}$ for a given model $f$, input $x$ and label $y$ is given as $\mathcal{L}(f(x), y) \stackrel{\text{def}}{=} \log(1 + \exp(-yf((\bar{w}, \bar{b}, \bar{a}), x)))$.

**Margin.** For data distribution $\mathcal{D}$, the margin of a model $f(x)$ is given as $\min_{(x,y) \sim \mathcal{D}} yf(x)$.

**Notation.** Here is some useful notation that we will use repeatedly. For a matrix $A$, $A(i,.)$ denotes the $i$th row of $A$. For any $k \in \mathbb{N}$, $\mathbb{S}^{k-1}$ denotes the surface of the unit norm Euclidean sphere in dimension $k$.

## 3.2 INITIALIZATIONS

The gradient descent dynamics of the network depends strongly on the scale of initialization. In this work, we primarily consider *rich regime* initialization.

**Rich regime.** In rich regime initialization, for any $i \in [m]$, the parameters $(\bar{w}(i,.), \bar{b}(i))$ of the first layer are sampled from a uniform distribution on $\mathbb{S}^d$. Each $\bar{a}(i)$ is sampled from $Unif\{-1,1\}$, and the output of the network is scaled down by $\frac{1}{m}$ (Chizat & Bach, 2020). This is roughly equivalent to Xavier initialization Glorot & Bengio (2010), where the weight parameters in both the layers are initialized approximately as $\mathcal{N}(0, \frac{2}{m})$ when $m \gg d$.

In addition, we also present some results for the lazy regime initialization described below.

**Lazy regime.** In the lazy regime, the weight parameters in the first layer are initialized with $\mathcal{N}(0, \frac{1}{d})$, those of second layer are initialized with $\mathcal{N}(0, \frac{1}{m})$ and the biases are initialized to 0 (Bietti & Mairal, 2019; Lee et al., 2019). This is approximately equivalent to Kaiming initialization (He et al., 2015).

## 3.3 THE INFINITE WIDTH LIMIT

For 1-hidden layer neural networks with ReLU activation in the infinite width limit i.e., as $m \to \infty$, Jacot et al. (2018); Chizat et al. (2019); Chizat & Bach (2020) gave interesting characterizations of the trained model. As mentioned above, the training process of these models falls into one of two regimes depending on the scale of initialization (Chizat et al., 2019):

**Rich regime.** In the infinite width limit, the neural network parameters can be thought of as a distribution $\nu$ over triples $(w, b, a) \in \mathbb{S}^{d+1}$ where $w \in \mathbb{R}^d, b, a \in \mathbb{R}$. Under the rich regime initialization, the function $f$ computed by the model can be expressed as

$$f(\nu, x) = \mathbb{E}_{(w,b,a) \sim \nu}[a(\phi(\langle w, x \rangle + b)]. \tag{1}$$

Chizat & Bach (2020) showed that the training process with rich initialization can be thought of as gradient flow on the Wasserstein-2 space and gave the following characterization of the trained model under the cross entropy loss $\mathbb{E}_{(x,y) \sim \mathcal{D}}[\mathcal{L}(\nu, (x, y))]$.

**Theorem 3.1.** *(Chizat & Bach, 2020) Under rich initialization in the infinite width limit with cross entropy loss, if gradient flow on 1-hidden layer NN with ReLU activation converges, it converges to a maximum margin classifier $\nu^*$ given as*

$$\nu^* = \arg\max_{\nu \in \mathcal{P}(\mathbb{S}^{d+1})} \min_{(x,y) \sim \mathcal{D}} yf(\nu, x), \tag{2}$$

*where $\mathcal{P}(\mathbb{S}^{d+1})$ denotes the space of distributions over $\mathbb{S}^{d+1}$.*

This training regime is known as the 'rich' regime since it learns data dependent features $\langle w, \cdot \rangle$.

**Lazy regime.** Jacot et al. (2018) showed that in the infinite width limit, the neural network behaves like a kernel machine. This kernel is popularly known as the Neural Tangent Kernel(NTK), and is given by $K(x, x') = \left\langle \frac{\partial f(x)}{\partial W}, \frac{\partial f(x')}{\partial W} \right\rangle$, where $W$ denotes the set of all trainable weight parameters. This initialization regime is called 'lazy' regime since the weights do not change much from initialization, and the NTK remains almost constant, i.e, the network does not learn data dependent features. We will use the following characterization of the NTK for 1-hidden layer neural networks.

**Theorem 3.2.** *Bietti & Mairal (2019) Under lazy regime initialization in the infinite width limit, the NTK for 1-hidden layer neural networks with ReLU activation i.e., $\phi(u) = \max(u, 0)$, is given as*

$$K(x, x') = \|x\| \|x'\| \kappa \left( \frac{\langle x, x' \rangle}{\|x\| \|x'\|} \right),$$

*where*

$$\kappa(u) = \frac{1}{\pi}(2u(\pi - cos^{-1}(u)) + \sqrt{1 - u^2}).$$

Figure 2: Illustration of an IFM dataset. Given a class $\pm 1$ represented by blue and red respectively, each coordinate value is drawn independently from the corresponding distribution. Shown above are the supports of distributions on three different coordinates for an illustrative IFM dataset, for positive and negative labels.

*Lazy regime for binary classification.* Soudry et al. (2018) showed that for linearly separable datasets, gradient descent for linear predictors on logistic loss converges to the max-margin support vector machine (SVM) classifier. This implies that, any sufficiently wide neural network, when trained for a finite time in the lazy regime on a dataset that is separable by the finite-width induced NTK, will tend towards the $\mathcal{L}_2$ max-margin-classifier given by

$$\underset{f \in \mathcal{H}}{\arg\min} \|f\|_{\mathcal{H}} \text{ s.t. } yf(x) \geq 1 \,\forall\, (x, y) \sim \mathcal{D}, \tag{3}$$

where $\mathcal{H}$ represents the Reproducing Kernel Hilbert Space (RKHS) associated with the finite width kernel (Chizat, 2020). With increasing width, this kernel tends towards the infinite-width NTK (which is universal (Ji et al., 2020)). Therefore, in lazy regime, we will focus on the $\mathcal{L}_2$ max-margin-classifier induced by the infinite-width NTK.

## 4 CHARACTERIZATION OF SB IN 1-HIDDEN LAYER NEURAL NETWORKS

In this section, we first theoretically characterize the SB exhibited by gradient descent on linearly separable datasets in the *independent features model (IFM)*. The main result, stated in Theorem 4.1, is that for binary classification of inputs in $\mathbb{R}^d$, even if there is a *single* coordinate in which the data is linearly separable, gradient descent dynamics will learn a model that relies *solely* on this coordinate, even when there are an arbitrarily large number $d-1$ of coordinates in which the data is separable, but by a non-linear classifier. In other words, the simplicity bias of these networks is characterized by *low dimensional input dependence*, which we denote by LD-SB. We then experimentally verify that NNs trained on some real datasets do indeed satisfy LD-SB.

### 4.1 DATASET

We consider datasets in the independent features model (IFM), where the joint distribution over $(x, y)$ satisfies $p(x, y) = r(y) \prod_{i=1}^{d} q_i(x_i|y)$, i.e, the features are distributed independently conditioned on the label $y$. Here $r(y)$ is a distribution over $\{-1, +1\}$ and $q_i(x_i|y)$ denotes the conditional distribution of $i^{\text{th}}$-coordinate $x_i$ given $y$. IFM is widely studied in literature, particularly in the context of naive-Bayes classifiers Lewis (1998). We make the following assumptions which posit that there are at least two features of differing complexity for classification: *one* with a linear boundary and *at least* one other with a non-linear boundary. See Figure 2 for an illustrative example.

- One of the coordinates (say, the 1$^{\text{st}}$ coordinate WLOG) is separable by a linear decision boundary with margin $\gamma$ (see Figure 2), i.e, $\exists \gamma > 0$, such that $\gamma \in Supp(q_1(x_1|y = +1)) \subseteq [\gamma, \infty)$ and $-\gamma \in Supp(q_1(x_1|y = -1)) \subseteq (-\infty, -\gamma]$, where $Supp(\cdot)$ denotes the support of a distribution.
- None of the other coordinates is linearly separable. More precisely, for all the other coordinates $i \in [d] \setminus \{1\}, 0 \in Supp(q_i(x_i|y = -1))$ and $\{-1, +1\} \subseteq Supp(q_i(x_i|y = +1))$.
- The dataset can be perfectly classified even without using the linear coordinate. This means, $\exists i \neq 1$, such that $q_i(x_i|y)$ has disjoint support for $y = +1$ and $y = -1$.

Though we assume axis aligned features, our results also hold for any rotation of the dataset. While our results hold in the general IFM setting, in comparison, current results for SB e.g., Shah et al. (2020), are obtained for *very specialized* datasets within IFM, and do not apply to IFM in general.

## 4.2 MAIN RESULT

Our main result states that, for rich initialization (Section 3.2), NNs demonstrate LD-SB for any IFM dataset satisfying the above conditions. Its proof appears in Appendix A.1.

**Theorem 4.1.** *For any dataset in the IFM model satisfying the above conditions and $\gamma \geq 1$, if gradient flow for 1-hidden layer FCN under rich initialization in the infinite width limit with cross entropy loss converges, it converges to $\nu^* = 0.5\delta_{\theta_1} + 0.5\delta_{\theta_2}$ on $\mathcal{S}^{d+1}$, where $\theta_1 = (\frac{\gamma}{\sqrt{2(1+\gamma^2)}}\mathbf{e}_1, \frac{1}{\sqrt{2(1+\gamma^2)}}, 1/\sqrt{2}), \theta_2 = (-\frac{\gamma}{\sqrt{2(1+\gamma^2)}}\mathbf{e}_1, \frac{1}{\sqrt{2(1+\gamma^2)}}, -1/\sqrt{2})$ and $\mathbf{e}_1 \stackrel{def}{=} [1, 0, \cdots, 0]$ denotes first standard basis vector. This implies $f(\nu^*, Px_1 + P_\perp x_2) = f(\nu^*, x_1)$ $\forall (x_1, y_1), (x_2, y_2) \sim \mathcal{D}$, where $P$ represents the (rank-1) projection matrix on first coordinate.*

Moreover, since at least one of the coordinates $\{2, \ldots, d\}$ has disjoint support for $q_i(x_i|y = +1)$ and $q_i(x_i|y = -1)$, $P_\perp(x)$ can still perfectly classify the given dataset, thereby implying LD-SB.

It is well known that the rich regime is more relevant for the practical performance of NNs since it allows for feature learning, while lazy regime does not (Chizat et al., 2019). Nevertheless, in the next section, we present theoretical evidence that LD-SB holds even in the lazy regime, by considering a much more specialized dataset within IFM.

## 4.3 LAZY REGIME

In this regime, we will work with the following dataset within the IFM family:

For $y \in \{\pm 1\}$ we generate $(x, y) \in D$ as

$$\mathbf{x}_1 = \gamma y$$

$$\forall i \in 2, .., d, \mathbf{x}_i = \begin{cases} \pm 1 & \text{for} & y = 1 \\ 0 & \text{for} & y = -1 \end{cases}$$

Although the dataset above is a point mass dataset, it still exhibits an important characteristic in common with the rich regime dataset – only one of the coordinates is linearly separable while others are not. For this dataset, we provide the characterization of max-margin NTK (as in Eqn. (3)):

**Theorem 4.2.** *For sufficiently small $\epsilon > 0$, there exists an absolute constant $N$ such that for all $d > N$ and $\gamma \in [7, \epsilon\sqrt{d})$, the $\mathcal{L}_2$ max-margin classifier for joint training of both the layers of 1-hidden layer FCN in the NTK regime on the dataset $D$, i.e., any $f$ satisfying Eqn. (3) satisfies:*

$$pred(f(Px_1 + P_\perp x_2)) = pred(f(x_1)) \forall (x_1, y_1), (x_2, y_2) \in D$$

*where $P$ represents the projection matrix on the first coordinate and $pred(f(x))$ represents the predicted label by the model $f$ on $x$.*

The above theorem shows that the prediction on a *mixed* example $Px_1 + P_\perp x_2$ is the same as that on $x_1$, thus establishing LD-SB. The proof for this theorem is provided in Appendix A.2.

## 4.4 EMPIRICAL VERIFICATION

In this section, we will present empirical results demonstrating LD-SB on 3 real datasets: Imagenette (FastAI, 2021), a binary version of Imagenette (b-Imagenette) and waterbirds-landbirds (Sagawa et al., 2020a) as well as one designed dataset MNIST-CIFAR (Shah et al., 2020). More details about the datasets can be found in Appendix B.1.

### 4.4.1 EXPERIMENTAL SETUP

We take Imagenet pretrained Resnet-50 models, with $2048$ features, for feature extraction and train a 1-hidden layer fully connected network, with ReLU nonlinearity, and $100$ hidden units, for classification on each of these datasets. During the finetuning process, we freeze the backbone Resnet-50 model and train only the 1-hidden layer head (more details in Appendix B.1) .

**Demonstrating LD-SB**: Given a model $f(\cdot)$, we establish its low dimensional SB by identifying a small dimensional subspace, identified by its projection matrix $P$, such that if we *mix* inputs $x_1$ and

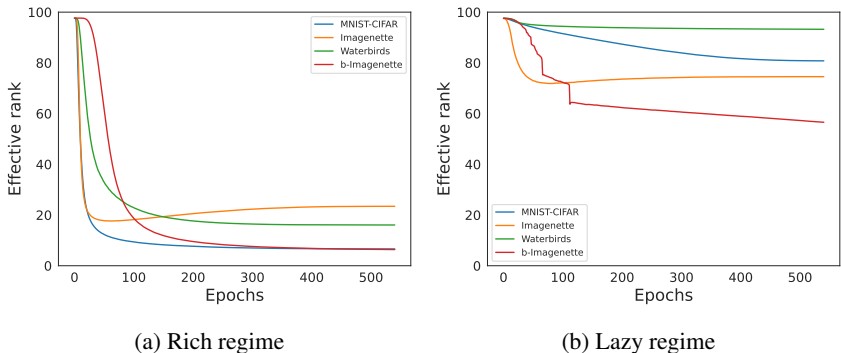

|               | (a) Rich regime | (b) Lazy regime |

Figure 3: Evolution of effective rank of first layer weight matrices in rich and lazy regimes.

Table 1: Demonstration of LD-SB in the rich regime: This table presents $P_\perp$ and $P$ randomized accuracies (RA) as well as logit changes (LC) on the four datasets. These results confirm that projection of input $x$ onto the subspace spanned by $P$ essentially determines the model's prediction on $x$. $\uparrow$ (resp. $\downarrow$) indicates that LD-SB implies a large (resp. small) value.

| Dataset | rank $(P)$ | Acc$(f)$ | $P_\perp$-RA ($\uparrow$) | $P$-RA ($\downarrow$) | $P_\perp$-LC ($\downarrow$) | $P$-LC ($\uparrow$) |
|---------|-----------|----------|---------------|-----------|------------|----------|
| b-Imagenette | 1 | 93.05±0.26 | 89.94±0.22 | 49.53±0.24 | 28.57±0.26 | 92.13 ± 0.24 |
| Imagenette | 10 | 79.52±0.13 | 75.89±0.25 | 9.33 ± 0.01 | 33.64±1.21 | 106.29±0.53 |
| Waterbirds | 3 | 91.88 ± 0.1 | 91.47±0.11 | 62.51±0.07 | 25.24±1.03 | 102.35±0.19 |
| MNIST-CIFAR | 1 | 99.69 ± 0.0 | 94.15±0.21 | 55.2 ± 0.13 | 38.97±0.76 | 101.98±0.31 |

$x_2$ as $Px_1 + P_\perp x_2$, the model's output on the mixed input $\widetilde{x} \stackrel{\text{def}}{=} Px_1 + P_\perp x_2$, $f(\widetilde{x})$ is always *close* to the model's output on $x_1$ i.e., $f(x_1)$. We measure *closeness* in four metrics: (1) $P_\perp$-randomized accuracy ($P_\perp$-RA): accuracy on the dataset $(Px_1 + P_\perp x_2, y_1)$ where $(x_1, y_1)$ and $(x_2, y_2)$ are sampled iid from the dataset, (2) $P$-randomized accuracy ($P$-RA): accuracy on the dataset $(Px_1 + P_\perp x_2, y_2)$, (3) $P_\perp$ logit change ($P_\perp$-LC): relative change wrt logits of $x_1$ i.e., $\|f(\widetilde{x}) - f(x_1)\| / \|f(x_1)\|$, and (4)$P$ logit change ($P$-LC): relative change wrt logits of $x_2$ i.e., $\|f(\widetilde{x}) - f(x_2)\| / \|f(x_2)\|$.

As described in Sections 4.2 and 4.3, the training of 1-hidden layer neural networks might follow different trajectories depending on the scale of initialization. So, the subspace projection matrix $P$ will be obtained in different ways for rich vs lazy regimes. For rich regime, we will empirically show that the first layer weights have a low rank structure as per Theorem 4.1 while for lazy regime, we will show that though first layer weights do not exhibit low rank structure, the model still has low dimensional dependence on the input as per Theorem 4.2.

### 4.4.2 RICH REGIME

Theorem 4.1 suggests that asymptotically, the first layer weight matrix will be low rank. However, since we train only for a finite amount of time, the weight matrix will only be approximately low rank. To quantify this, we use the notion of effective rank Roy & Vetterli (2007) to measure the rank of the first layer weight matrix.

**Definition 4.3.** *Given a matrix $M$, its effective rank is defined as:* $\text{Eff-rank}(M) = e^{-\sum_i \overline{\sigma_i(M)^2} \log \overline{\sigma_i(M)^2}}$ *where $\sigma_i(M)$ denotes the $i^{th}$ singular value of $M$ and $\overline{\sigma_i(M)^2} \stackrel{\text{def}}{=} \frac{\sigma_i(M)^2}{\sum_i \sigma_i(M)^2}$.*

One way to interpret the effective rank is that it is the exponential of von-Neumann entropy Petz (2001) of the matrix $\frac{MM^\top}{\text{Tr}(MM^\top)}$, where $\text{Tr}(\cdot)$ denotes the trace of a matrix. For illustration, the effective rank of a projection matrix onto $k$ dimensions equals $k$.

Figure 3a shows the evolution of the effective rank through training on the four datasets. We observe that the effective rank of the weight matrix decreases drastically towards the end of training. To confirm that this indeed leads to LD-SB, we set $P$ to be the subspace spanned by the top singular

Table 2: Demonstration of LD-SB in the lazy regime: This table presents $P_\perp$ and $P$ randomized accuracies as well as logit changes on the four datasets. These results confirm that the projection of input $x$ onto the subspace spanned by $P$ essentially determines the model's prediction on $x$.

| Dataset | rank $(P)$ | Acc($f$) | $P_\perp$-RA ($\uparrow$) | $P$-RA ($\downarrow$) | $P_\perp$-LC ($\downarrow$) | $P$-LC ($\uparrow$) |
|---|---|---|---|---|---|---|
| b-Imagenette | 1 | 92.75±0.06 | 90.07±0.34 | 52.09±1.34 | 36.94±1.01 | 138.41±1.62 |
| Imagenette | 15 | 79.97±0.44 | 68.25±1.18 | 11.92±0.82 | 55.99±3.86 | 133.86±5.42 |
| Waterbirds | 6 | 90.46±0.07 | 89.67±0.42 | 62.44±4.48 | 36.89±5.18 | 105.41±7.06 |
| MNIST-CIFAR | 2 | 99.74 ± 0.0 | 99.45±0.17 | 49.83±0.67 | 24.9 ± 0.61 | 141.12±1.86 |

Table 3: Mistake diversity and class conditioned logit correlation of models trained independently (Mist-Div $(f, f_{\text{ind}})$ and CC-LogitCorr $(f, f_{\text{ind}})$ resp.) vs trained sequentially after projecting out important features of the first model (Mist-Div $(f, f_{\text{proj}})$ and CC-LogitCorr $(f, f_{\text{proj}})$ resp.). The results demonstrate that $f$ and $f_{\text{proj}}$ are more diverse compared to $f$ and $f_{\text{ind}}$.

| Dataset | Mist-Div $(f, f_{\text{ind}})$ ($\uparrow$) | Mist-Div $(f, f_{\text{proj}})$ ($\uparrow$) | CC-LogitCorr $(f, f_{\text{ind}})$ ($\downarrow$) | CC-LogitCorr $(f, f_{\text{proj}})$ ($\downarrow$) |
|---|---|---|---|---|
| B-Imagenette | 3.87 ± 1.54 | 21.15 ± 1.57 | 99.88 ± 0.01 | 90.86 ± 1.08 |
| Imagenette | 6.6 ± 0.46 | 11.44 ± 0.65 | 99.31 ± 0.12 | 91 ± 0.59 |
| Waterbirds | 2.9 ± 0.52 | 14.53 ± 0.48 | 99.66 ± 0.04 | 93.81 ± 0.48 |
| MNIST-CIFAR | 0.0 ± 0.0 | 5.56 ± 7.89 | 99.76 ± 0.17 | 78.74 ± 2.28 |

directions of the first layer weight matrix and compute $P$ and $P_\perp$ randomized accuracies as well as the relative logit change. The results, presented in Table 1 confirm LD-SB in the rich regime on these datasets.

### 4.4.3 LAZY REGIME

For the lazy regime, it turns out that the rank of first layer weight matrix remains high throughout training, as shown in Figure 3b. However, we are able to find a low dimensional projection matrix $P$ satisfying the conditions of LD-SB (as stated in Def 1.1) as the solution to an optimization problem. More concretely, given a pretrained model $f$ and a rank $r$, we obtain a *projection matrix $P$* solving:

$$\min_P \frac{1}{n} \sum_{i=1}^{n} \left( \mathcal{L}\left( f(Px_i), y_i \right) + \lambda \mathcal{L}\left( f(P^\perp x_i), \mathcal{U}[L] \right) \right)$$

where $\mathcal{U}[L]$ represents a uniform distribution over all the $L$ labels, $(x_1, y_1), \cdots, (x_n, y_n)$ are training examples and $\mathcal{L}(\cdot, \cdot)$ is the cross entropy loss. We reiterate that the optimization is only over $P$, while the model parameters $f$ are unchanged. In words, the above function ensures that the neural network produces correct predictions along $P$ and uninformative predictions along $P_\perp$. Table 2 presents the results for $P_\perp$ and $P$-RA as well as LC. As can be seen, even in this case, we are able to find small rank projection matrices demonstrating LD-SB.

## 5 TRAINING DIVERSE CLASSIFIERS USING *OrthoP*

Motivated by our results on low dimensional SB, in this section, we present a natural way to train diverse models, so that an ensemble of such models could mitigate SB. More concretely, given an initial model $f$ with a low dimensional projection $P$ that captures its input dependence, we train another model $f_{\text{proj}}$ by projecting the input through $P_\perp$ i.e., instead of using dataset $(x_i, y_i)$ for training, we use $(P_\perp x_i, y_i)$ for training the second model (denoted by $f_{\text{proj}}$). We refer to this training procedure as $OrthoP$ for *orthogonal projection*.

Given any two models $f$ and $\tilde{f}$, we evaluate their diversity using two metrics. The first is mistake diversity: Mist-Div $\left( f, \tilde{f} \right) \stackrel{\text{def}}{=} 1 - \frac{|\{i : f(\mathbf{x}_i) \neq y_i \ \& \ \tilde{f}(\mathbf{x}_i) \neq y_i\}|}{\min(|\{i : f(\mathbf{x}_i) \neq y_i\}|, |\{i : \tilde{f}(\mathbf{x}_i) \neq y_i\}|)}$, where we abuse notation by using

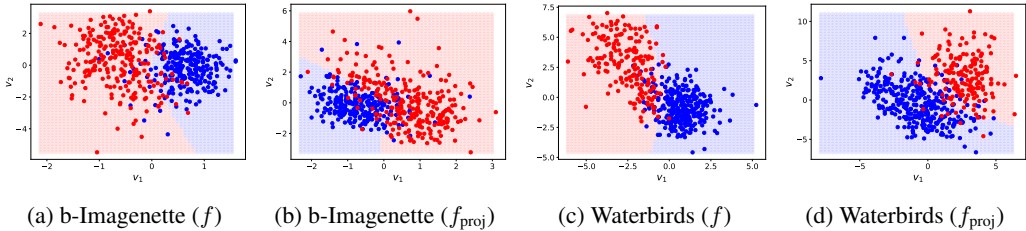

(a) b-Imagenette ($f$)    (b) b-Imagenette ($f_{\text{proj}}$)    (c) Waterbirds ($f$)    (d) Waterbirds ($f_{\text{proj}}$)

Figure 4: Decision boundaries for $f$ and $f_{\text{proj}}$ for B-Imagenette and Waterbirds datasets, visualized in the top 2 singular directions of the first layer weight matrix. The decision boundary of $f_{\text{proj}}$ is more non-linear compared to that of $f$.

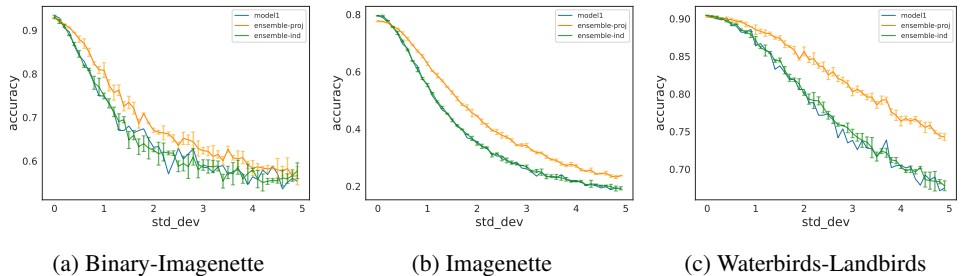

(a) Binary-Imagenette    (b) Imagenette    (c) Waterbirds-Landbirds

Figure 5: Variation of test accuracy vs standard deviation of Gaussian noise added to the pretrained representations of the dataset. Model 1 (i.e., $f$) is kept fixed, and values for both the ensembles are averaged across 3 runs. Standard deviation is shown by the error bars.

$f(x_i)$ (resp. $\tilde{f}(x_i)$) to denote the class predicted by $f$ (resp $\tilde{f}$) on $x_i$. Higher Mist-Div $\left(f, \tilde{f}\right)$ means that there is very little overlap in the mistakes of $f$ and $\tilde{f}$. The second is class conditioned logit correlation i.e., correlation between outputs of $f$ and $\tilde{f}$, conditioned on the class. More concretely, CC-LogitCorr $\left(f, \tilde{f}\right) = \frac{\sum_{y \in \mathcal{Y}} \text{Corr}\left([f(\mathbf{x}_i)], [\tilde{f}(\mathbf{x}_i)] : y_i = y\right)}{|\mathcal{Y}|}$, where $\text{corr}([f(\mathbf{x}_i)], [\tilde{f}(\mathbf{x}_i)] : y_i = y)$ represents the empirical correlation between the logits of $f$ and $\tilde{f}$ on the data points where the true label is $y$. Table 3 compares the diversity of two independently trained models ($f$ and $f_{\text{ind}}$) with that of two sequentially trained models ($f$ and $f_{\text{proj}}$) as above. The results demonstrate that $f$ and $f_{\text{proj}}$ are more diverse compared to $f$ and $f_{\text{ind}}$. Figure 4 shows the decision boundary of $f$ and $f_{\text{proj}}$ on 2-dimensional subspace spanned by top two singular vectors of the weight matrix. We observe that the decision boundary of the second model is more non-linear compared to that of the first model.

Finally, Figure 5 shows the variation of test accuracy with the strength of gaussian noise added to the pretrained representations of the dataset. We can see that an ensemble of $f$ and $f_{\text{proj}}$ is much more robust as compared to an ensemble of $f$ and $f_{\text{ind}}$ (where an ensemble is obtained by averaging the logits of the two models).

## 6 DISCUSSION

In this work, we characterize the simplicity bias exhibited by one hidden layer neural networks in terms of the low-dimensional input dependence of the model. We provide a theoretical proof for linearly separable datasets, and validate it empirically on real datasets. Based on this characterization, we also propose a simple way to train diverse models and show that it leads to models with significantly better Gaussian noise robustness.

This work is an initial step towards rigorously defining simplicity bias or shortcut learning of neural networks, which is one of the major challenges to their real-world deployment (Geirhos et al., 2020). Providing a similar characterization for deeper nets and other architectures is an important research direction, which, in our opinion, requires a better understanding of the training dynamics and limit points of gradient descent on these networks.

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

# A  PROOFS FOR RICH AND LAZY REGIME

## A.1  RICH REGIME

We restate Theorem 4.1 below and prove it.

**Theorem A.1.** *For any dataset in IFM model satisfying the conditions in Section 4.1, $\gamma \geq 1$ and $f(\nu, x)$ as in Eqn. (1), the distribution $\nu^* = 0.5\delta_{\theta_1} + 0.5\delta_{\theta_2}$ on $\mathcal{S}^{d+1}$ is the unique max-margin classifier satisfying Eqn. (2), where $\theta_1 = (\frac{\gamma}{\sqrt{2(1+\gamma^2)}}\mathbf{e}_1, \frac{1}{\sqrt{2(1+\gamma^2)}}, 1/\sqrt{2}), \theta_2 = (-\frac{\gamma}{\sqrt{2(1+\gamma^2)}}\mathbf{e}_1, \frac{1}{\sqrt{2(1+\gamma^2)}}, -1/\sqrt{2})$ and $\mathbf{e}_1 \overset{def}{=} [1, 0, \cdots, 0]$ denotes first standard basis vector. In particular, this implies that if gradient flow for 1-hidden layer FCN under rich initialization in the infinite width limit with cross entropy loss converges, it converges to $\nu^*$ satisfying $f(\nu^*, Px_1 + P_\perp x_2) = f(\nu^*, x_1)\forall(x_1, y_1), (x_2, y_2) \in D$, where $P$ represents the (rank-1) projection matrix on the first coordinate.*

*Proof of Theorem A.1:*  (Chizat & Bach, 2020) showed the following primal-dual characterization of maximum margin classifiers in eqn. (2):

**Lemma A.2.** *(Chizat & Bach, 2020) $\nu^*$ satisfies eqn. (2) if there exists a data distribution $p^*$ such that the following two complementary slackness conditions hold:*

$$Supp(\nu^*) \subseteq \underset{(w,b,a)\in\mathbb{S}^{d+1}}{\arg\max} \ \mathbb{E}_{(x,y)\sim p^*}y[a(\phi(\langle w, x \rangle + b))] \quad and \tag{4}$$

$$Supp(p^*) \subseteq \underset{(x,y)\sim\mathcal{D}}{\arg\min} y\mathbb{E}_{(w,b,a)\sim\nu^*}[a(\phi(\langle w, x \rangle + b))]. \tag{5}$$

The plan is to construct a distribution $p^*$ that satisfies the conditions of the above Lemma.

**Uniqueness.**  Note further that for a fixed $p^*$, $\mathbb{E}_{(x,y)\sim p^*}yf(\nu, x)$ is an upper bound for the margin $\min_{(x,y)\sim\mathcal{D}} yf(\nu, x)$ of any classifier $\nu$. Hence, for uniqueness, it suffices to show that $\delta_{\theta_1}, \delta_{\theta_2}$ are the unique maximizers of the objective on the RHS of eqn. (4) and that the unique maximum margin convex combination of $\delta_{\theta_1}, \delta_{\theta_2}$ over $\mathcal{D}$ is $\nu^*$.

We first describe the support $D$ of $p^*$. For $y \in \{\pm 1\}$ we generate $(x, y) \in D$ as

$$\mathbf{x}_1 = \gamma y$$

$$\forall i \in 2, .., d, \mathbf{x}_i = \begin{cases} \pm 1 & for \quad y = 1 \\ 0 & for \quad y = -1 \end{cases}$$

Now for $(x, y) \in D$, define

$$p^*(x, y) = \begin{cases} 0.5 & for \quad y = 1 \\ 0.5^d & for \quad y = -1 \end{cases} \tag{6}$$

Note that $p^*$ is supported on $2^{d-1}$ positive instances and one negative instance. We begin by showing eqn. (5).

**Claim A.3.** *$p^*$ as in eqn. (6) satisfies eqn. (5). Further, the unique maximum margin convex combination of $\delta_{\theta_1}, \delta_{\theta_2}$ is $\nu^*$.*

*Proof.*  Let us find the minimizers $(x, y) \sim \mathcal{D}$ of $yf(\nu, x) = y\mathbb{E}_{(w,b,a)\sim\nu^*}[a(\phi(\langle w, x \rangle + b))]$ for any $\nu = \lambda\delta_{\theta_1} + (1-\lambda)\delta_{\theta_2}, 0 \leq \lambda \leq 1$.

$yf(\nu, x)$ for $(x, y)$ with $y = -1$ (denoting $x_1$ by $-\alpha_1$, where $\alpha_1 \geq \gamma$) is

$$yf(\nu, x) = -1\Big[\lambda * \phi\left(\frac{\gamma}{\sqrt{2(1+\gamma^2)}}\mathbf{e}_1^\top(-\alpha_1\mathbf{e}_1) + \frac{1}{\sqrt{2(1+\gamma^2)}}\right) * \frac{1}{\sqrt{2}}$$

$$+ (1-\lambda) * \phi\left(-\frac{\gamma}{\sqrt{2(1+\gamma^2)}}\mathbf{e}_1^\top(-\alpha_1\mathbf{e}_1) + \frac{1}{\sqrt{2(1+\gamma^2)}}\right) * \frac{-1}{\sqrt{2}}\Big],$$

and for $(x, y)$ with $y = 1$ (denoting $x_1$ by $\alpha_2$, where $\alpha_2 \geq \gamma$) is

$$
yf(\nu, x) = 1\Big[\lambda * \phi\left(\frac{\gamma}{\sqrt{2(1+\gamma^2)}}\mathbf{e}_1^\top(\alpha_2\mathbf{e}_1) + \frac{1}{\sqrt{2(1+\gamma^2)}}\right) * \frac{1}{\sqrt{2}}
$$
$$
+ (1 - \lambda) * \phi\left(-\frac{\gamma}{\sqrt{2(1+\gamma^2)}}\mathbf{e}_1^\top(\alpha_2\mathbf{e}_1) + \frac{1}{\sqrt{2(1+\gamma^2)}}\right) * \frac{-1}{\sqrt{2}}\Big].
$$

As $\gamma \geq 1$, the expressions above equal $\lambda\frac{\sqrt{\gamma\alpha_1+1}}{2}$ and $(1 - \lambda)\frac{\sqrt{\gamma\alpha_2+1}}{2}$ respectively, and hence are minimized at $\alpha_1 = \alpha_2 = \gamma$. Hence, the margin of $\nu$ is $\min(\lambda, 1 - \lambda)\frac{\sqrt{1+\gamma^2}}{2}$ which is uniquely maximized at $\lambda = 1/2$. Further for $\lambda = 1/2$, all points in $D$ have the same value of $yf(\nu, x)$. $\qquad\square$

In the rest of the proof we show eqn. (4), Let us denote by $g(w, b, a) := \mathbb{E}_{(x,y)\sim p^*}y[a(\phi(\langle w, x\rangle+b))]$. We show that $\delta_{\theta_1}, \delta_{\theta_2}$ are the only maximizers of $g(w, b, a)$ over $\mathbb{S}^{d+1}$.

We first find $g(\theta_1), g(\theta_2)$.

$$
g(\theta_1) = \Pr(y = 1) \cdot 1 \cdot \frac{1}{\sqrt{2}} \cdot \phi\left(\frac{\gamma}{\sqrt{2(1+\gamma^2)}}\mathbf{e}_1^T(\gamma\mathbf{e}_1) + \frac{1}{\sqrt{2(1+\gamma^2)}}\right)
$$
$$
+ \Pr(y = -1) \cdot -1 \cdot \frac{1}{\sqrt{2}} \cdot \phi\left(\frac{\gamma}{\sqrt{2(1+\gamma^2)}}\mathbf{e}_1^T(-\gamma\mathbf{e}_1) + \frac{1}{\sqrt{2(1+\gamma^2)}}\right) = \frac{\sqrt{\gamma^2 + 1}}{4},
$$

where the first term is because $w_2, w_3, \ldots, w_d$ are zero for $\theta_1$. Similarly, $g(\theta_2) = \frac{\sqrt{\gamma^2+1}}{4}$. We now show that $g(w, a, b) < \frac{\sqrt{\gamma^2+1}}{4}$ for $(w, a, b) \notin \{\theta_1, \theta_2\}$.

We begin by showing the following simple but useful claim.

**Claim A.4.** *All maximizers of $g(w, b, a)$ over $\mathbb{S}^{d+1}$ satisfy $|a| = 1/\sqrt{2}$.*

*Proof.* The proof essentially follows from the $1-$homogeneity of the ReLU function $\phi$ and separability of $g(w, b, a)$. Note that $g(w, b, a) = \sqrt{\|w\|^2 + b^2}a \cdot g(w', b', 1)$ where $\|w'\|^2 + b^2 = 1$. Maximizing $g(w, b, a)$ is equivalent to maximizing $g(w', b', 1)$ over $\mathbb{S}^d$ and $a\sqrt{\|w\|^2 + b^2}$ over $\mathbb{S}^{d+1}$ respectively. The second of these has its unique maximum at $|a| = 1/\sqrt{2}$, completing the proof. $\qquad\square$

Now express $g(w, b, a)$ as

$$
g(w, b, a) = a\left(\Pr(y = 1)\mathbb{E}[\phi(w^T x + b)|y = 1] - \Pr(y = -1)\mathbb{E}[\phi(w^T x + b)|y = -1]\right)
$$
$$
= \frac{a}{2}\left(\mathbb{E}_\sigma\left[\phi(\gamma w_1 + b + \sum_{i=2}^d \sigma_i w_i)\right] - \phi(b - \gamma w_1)\right), \tag{7}
$$

where $\sigma_i$ are independent Rademacher random variables. We have two cases on $a$:

**Case 1:** $a = 1/\sqrt{2}$. By eqn. (7) we have

$$
g(w, b, 1/\sqrt{2}) \leq \frac{1}{2\sqrt{2}}\mathbb{E}_\sigma\left[\phi(\gamma w_1 + b + \sum_{i=2}^d \sigma_i w_i)\right].
$$

To simplify the above, define the random variable $X = \sum_{i=2}^d \sigma_i w_i$ and denote $\gamma w_1 + b$ by $\alpha$. Note that $|\alpha| = |\gamma w_1 + b| \leq \sqrt{\frac{\gamma^2+1}{2}}$ which follows from $\|w\|^2 + b^2 = 1/2$. The expectation in the last expression above becomes

$$
\mathbb{E}[\phi(X + \alpha)] = \mathbb{E}[(X + \alpha)\mathbf{1}\{X + \alpha \geq 0\}] = \mathbb{E}[X\mathbf{1}\{X \geq -\alpha\}] + \alpha\Pr(X \geq -\alpha)
$$
$$
= \mathbb{E}[X\mathbf{1}\{X \geq \alpha\}] + \alpha(1 - \Pr(X \geq \alpha)) \leq \mathbb{E}[X\mathbf{1}\{X \geq \alpha\}] + \alpha,
$$

where the last equality follows from symmetry of $X$. Note that $\text{Var}(X) = \sum_{i=2}^{d} w_i^2$ which is at most $\frac{1}{2} - \frac{\alpha^2}{1+\gamma^2}$ (using $\gamma w_1 + b = \alpha$ and $\|w\|^2 + b^2 = 1/2$). Using A.5 to upper bound $\mathbb{E}[X\mathbf{1}\{X \geq \alpha\}]$ we have

$$\mathbb{E}[\phi(X + \alpha)] \leq \alpha + \sqrt{\frac{1}{2} \min\left(\frac{1}{2}, \frac{\frac{1}{2} - \frac{\alpha^2}{1+\gamma^2}}{2\alpha^2}\right) \left(\frac{1}{2} - \frac{\alpha^2}{1+\gamma^2}\right)}.$$

We can check that the RHS of the above has its unique maximizer at $\alpha = \sqrt{\frac{1+\gamma^2}{2}}$ for $|\alpha| \leq \sqrt{\frac{1+\gamma^2}{2}}$. Hence $g(w, b, a) \leq \frac{\sqrt{1+\gamma^2}}{4}$ in this case. We are now done since any $(w_1, b)$ satisfying $\gamma w_1 + b = \sqrt{\frac{1+\gamma^2}{2}}$ and $w_1^2 + b^2 \leq 1/2$ has $b = \frac{1}{\sqrt{2(1+\gamma^2)}}$.

**Case 2:** $a = -1/\sqrt{2}$. Using eqn. (7) we have $g(w, b, -1/\sqrt{2}) \leq \phi(b - \gamma w_1)/2\sqrt{2}$ which for $b^2 + w_1^2 \leq 1/2$ attains its unique maximum $\sqrt{\frac{\gamma^2+1}{4}}$ at $b = \frac{1}{\sqrt{2(1+\gamma^2)}}$.

Finally, note that the weights of the *trained* network $(w, b, a)$ are sampled from $\nu^*$. Hence, the final claim in the theorem about $f(\nu^*, Px_1 + P_\perp x_2)$ follows since the distribution of $w$ only has a support on $\mathbf{e}_1$ and $-\mathbf{e}_1$.

$\square$

### A.1.1 AUXILIARY LEMMAS FOR RICH REGIME

**Lemma A.5.** *For any symmetric discrete random variable $X$ with bounded variance, for $\alpha > 0$,*

$$\mathbb{E}[X\mathbb{I}(X \geq \alpha)] \leq \sqrt{\frac{1}{2} \min\left(\frac{1}{2}, \frac{Var(X)}{2\alpha^2}\right) Var(X)}.$$

*Proof.*

$$\mathbb{E}[X\mathbb{I}(X \geq \alpha)] = \sum_{x \geq \alpha} xp(x) = \sum_{x \geq \alpha} \sqrt{p(x)}\sqrt{p(x)}x \leq \sqrt{p(X \geq \alpha) \sum_{x \geq \alpha} x^2 p(x)}, \qquad (8)$$

where the last inequality is by Cauchy-Schwartz. Also by Chebyshev's inequality, $p(|X| \geq \alpha) \leq Var(X)/2\alpha^2$. Combining this with eqn. (8) and using symmetry of $X$ and non-negativity of $\alpha$ gives the required lemma. $\square$

### A.2 LAZY REGIME

Theorem 4.2 is a corollary of the following more general theorem.

**Theorem A.6.** *Consider a point $x \in D$. For sufficiently small $\epsilon > 0$, there exist an absolute constant $N$ such that for all $d > N, \gamma < \epsilon\sqrt{d}$ and $\gamma \geq 7$, for the joint training of both the layers of 1-hidden layer FCN in the NTK regime, the prediction of any point of the form $(\zeta, x_{2:d})$ satisfies the following:*

    *1. For $\zeta \geq 0.73$, the prediction is positive.*

    *2. For $\zeta \leq -0.95\gamma$, the prediction is negative.*

The above theorem establishes that perturbing $x_1$ by $O(\gamma)$ changes $pred(f(x))$ for $x \in D$ (whereas a classifier exists that achieves a margin of $\Omega(\sqrt{d})$ on $D$, as $D$ has margin 1 for coordinates $\{2 \cdots d\}$). As $\gamma = o(d)$, this shows that the learned model is adversarially vulnerable.

*Proof of Theorem A.6.* The idea of the proof is to obtain an explicit expression for $f(x)$ by applying standard kernel max-margin SVM theory to the NTK kernel 3.2.

We begin with some preliminaries. We will refer to the first coordinate of the instance as the 'linear' coordinate, and to the rest as 'non-linear' coordinates. Also, henceforth we append an extra coordinate with value 1 to all our instances (corresponding to bias term) - as is standard for working with unbiased SVM without loss of generality.

**Explicit expression for $f$.** Using representer theorem for max margin kernel SVM, we know that $f$ can be expressed as

$$f(x) = \sum_{(x^{(t)}, y^{(t)}) \in D} \lambda_t y^{(t)} K(x, x^{(t)}),$$

for some $\lambda_t \geq 0$ (that are known as *Lagrange multipliers*). Further by KKT conditions, a function possessing such a representation (that correctly classifies $D$) has maximum margin if $y^{(t)} f(x^{(t)}) = 1$ whenever $\lambda_t > 0$ (training points $t$ satisfying $\lambda_t > 0$ are called *support vectors*).

We begin with a useful claim.

**Claim A.7.** *The max margin kernel SVM for $D$ with the NTK kernel has all points in $D$ as support vectors.*

*Proof.* By the above discussion, it suffices to show that the (unique) solution $\alpha \in \mathbb{R}^{|D|}$ to $K\alpha = y$ satisfies $\text{sign}(\alpha_i) = y^{(i)}$ for all $i$, where $K$ is the $|D| \times |D|$ Gram matrix with $(i,j)$th entry $K(x^{(i)}, x^{(j)})$ and $y_i = y^{(i)}$ (the Lagrange multipliers $\lambda_i$ are then given by $y_i \alpha_i$).

*Structure of Gram matrix.* Order $D$ so that the positive instances appear first. Then the Gram matrix $K$ has a block structure of the form $\begin{pmatrix} B & C \\ C^T & R \end{pmatrix}$ where $B \in \mathbb{R}^{2^{d-1} \times 2^{d-1}}$ and $R \in \mathbb{R}$ are the Gram matrices for the positive and negative instances respectively, and $C \in \mathbb{R}^{2^{d-1} \times 1}$ represents the $K(x^{(i)}, x^{(|D|)})$ values for $i < |D|$.

Recall that for the NTK kernel, $K(x^{(i)}, x^{(j)})$ has the form $\|x^{(i)}\| \|x^{(j)}\| \kappa(\langle x^{(i)}, x^{(j)} \rangle)$. Note all the positive instances have the same norm (denoted by $\rho_1 = \sqrt{d + \gamma^2}$) and the inner product between two positive instances depends only on the number $i$ of non-matching non-linear coordinates (denoted by $\beta_i$ for $0 \leq i \leq d-1$). Hence, the rows of $B$ are permutations of each other, with the entry $\rho_1^2 \beta_i$ appearing $\binom{d-1}{i}$ times. Similarly, the entries in $C$ are all equal and are denoted by $\rho_1 \rho_2 \beta_d$ where $\beta_d$ denotes $\kappa(x^{(t)}, x^{|D|})$ for any $t < |D|$ and $\rho_2 = \|x^{|D|}\| = \sqrt{1 + \gamma^2}$. The only entry in $R$ is $\rho_2^2 \kappa(1)$. In particular,

$$\beta_i = \kappa\left(\frac{d - 2i + \gamma^2}{d + \gamma^2}\right) \text{ for } i \in [|D| - 1], \quad \text{and} \quad \beta_d = \kappa\left(\frac{1 - \gamma^2}{\sqrt{d + \gamma^2}\sqrt{1 + \gamma^2}}\right).$$

Now we are ready to solve $K\alpha = y$. By symmetry in the structure of K, $\alpha$ looks like $[a, a, \ldots\ldots, b]$, where the first $|D| - 1$ entries are the same.

Expanding $K\alpha = y$, we get two equations given by

$$a\rho_1^2 \left( \sum_{i=0}^{d-1} \binom{d-1}{i} \beta_i \right) + b\rho_1 \rho_2 \beta_d = 1 \quad \text{and} \quad 2^{d-1} a \rho_1 \rho_2 \beta_d + \rho_2^2 \kappa(1) b = -1.$$

Solving, we get

$$a = \frac{\rho_2 \kappa(1) + \rho_1 \beta_d}{\rho_1^2 \rho_2 \sum_{i=0}^{d-1} \left( \binom{d-1}{i} [\kappa(1)\beta_i - \beta_d^2] \right)} \quad \text{and} \quad b = \frac{-1 - 2^{d-1} a \rho_1 \rho_2 \beta_d}{\rho_2^2 \kappa(1)}.$$

We now show that $a > 0$ and $b < 0$. Note that for sufficiently large $d$, $\beta_d$ can be made arbitrarily close to $\kappa(0) = 1/\pi$ (since $\kappa$ is smooth around 0). Hence, $a > 0$ implies $b < 0$. We in fact give the following estimate for $a$:

$$a = 2^{1-d} \cdot \frac{\rho_2 \kappa(1) + \rho_1 \beta_d}{\xi \rho_1^2 \rho_2} \quad \text{where} \quad \frac{2}{\pi} - \frac{1}{\pi^2} + O\left(\frac{1}{d}\right) \leq \xi \leq 2 + O\left(\frac{1}{d}\right). \quad (9)$$

For the lower bound on $\xi$, write

$$\sum_{i=0}^{d-1} \binom{d-1}{i} [\kappa(1)\beta_i - \beta_d^2] = \kappa(1) \sum_{i=0}^{\lfloor d/2 \rfloor} \binom{d-1}{i} (\beta_i + \beta_{d-1}) - 2^{d-1}\beta_d^2$$

$$\geq \kappa(1) \sum_{i=0}^{\lfloor d/2 \rfloor} \binom{d-1}{i} 2\beta_{d/2} - 2^{d-1}\beta_d^2 \geq 2^{d-1} \left( \kappa(1)\kappa(0) - \kappa^2(0) + O\left(\frac{1}{d}\right) \right),$$

where for the first inequality we used convexity of $\kappa$ and for the second inequality we used $\beta_{d/2} = \kappa(0) + O(1/d), \beta_d = \kappa(0) + O(1/\sqrt{d})$. For the upper bound on $\xi$, write

$$\sum_{i=0}^{d-1} \binom{d-1}{i} [\kappa(1)\beta_i - \beta_d^2] \leq \kappa(1) \sum_{i=0}^{d-1} \binom{d-1}{i} \kappa\left(1 - \frac{2i}{d+\gamma^2}\right)$$

$$\leq \kappa(1) \sum_{i=0}^{d-1} \binom{d-1}{i} \left(2 - \frac{2i}{d+\gamma^2}\right) = \kappa(1)2^d - \frac{\kappa(1)(d-1)2^{d-1}}{d+\gamma^2},$$

where for the second inequality we used $\kappa(u) \leq 1 + u$ (which holds by convexity and $\kappa(-1) = 0, \kappa(1) = 2$). $\qquad\square$

Now we analyze predicted labels for points of the form $(\zeta, x_{2:d+1})$ where $x \in D$. We make two cases depending on the label of $x$.

**Predicted label for point $(\zeta, x_{2:d+1}^{(t)})$ where $x^{(t)} \in D$ has positive label**

Our point (denoted by $x$) has the form $(\zeta, \zeta_1, \zeta_2, \ldots, \zeta_d, 1)$ where $\zeta_i \in \pm 1$. The idea of the proof is to write $f$ explicitly as a function of $\zeta$ and work with its first order Taylor expansion around $\zeta = \gamma$, with some additional work to take care of non-smoothness of $f$.

*Explicit form for $f$.* Let $\tau_i \overset{\text{def}}{=} \langle x, x' \rangle / (\|x\|\|x'\|)$ for a positive instance $x' \in D$, where $x$ and $x'$ have exactly $i$ non-matching non-linear coordinates (for $0 \leq i \leq d-1$). Similarly denote by $\tau_d$ the quantity $\langle x, x^{|D|} \rangle / (\|x\|\|x^{|D|}\|)$. In particular,

$$\tau_i = \left(\frac{d - 2i + \gamma\zeta}{\rho_1\|x\|}\right) \qquad \text{and} \qquad \tau_d = \left(\frac{1 - \gamma\zeta}{\rho_2\|x\|}\right).$$

By the above discussion, we have

$$f(x) = a\left(\sum_{t=1}^{|D|-1} K(x, x^{(t)})\right) + bK(x, x^{|D|}) = a\rho_1\|x\| \left(\sum_{i=0}^{d-1} \binom{d-1}{i}\kappa(\tau_i)\right) + b\rho_2\|x\|\kappa(\tau_d).$$

Substituting $b$ and denoting $f(x)/\|x\|$ by $g(\zeta)$ we get

$$g(\zeta) = a\rho_1 \left[\sum_{i=0}^{d-1} \binom{d-1}{i}\kappa(\tau_i(\zeta)) - \frac{2^{d-1}\beta_d}{\kappa(1)}\kappa(\tau_d(\zeta))\right] - \frac{\kappa(\tau_d(\zeta))}{\rho_2\kappa(1)}. \tag{10}$$

Now try to expand $g(\zeta)$ using the Taylor series around $\zeta = \gamma$ (note that $g(\gamma) = 1/\rho_1$). Note that $\kappa'$ can however be unbounded around $-1$ and $1$. To get around this, write $g = h + q$, where $h$ has bounded first and second derivative, and $q$ has lower order than $h$ for $\zeta$ of interest. In particular,

$$h(\zeta) = a\rho_1 \left[\sum_{i=d/4}^{3d/4} \binom{d-1}{i}\kappa(\tau_i(\zeta)) - \frac{2^{d-1}\beta_d}{\kappa(1)}\kappa(\tau_d(\zeta))\right] - \frac{\kappa(\tau_d(\zeta))}{\rho_2\kappa(1)} \qquad \text{and}$$

$$q(\zeta) = a\rho_1 \left[\sum_{i:|d/2-i|>d/4} \binom{d-1}{i}\kappa(\tau_i(\zeta))\right].$$

Observe that $q(\zeta) = o(c^d)$ for $c < 1$ using the estimate eqn. (9) for $a$ and concentration for sums of independent Bernoullis. By Taylor's theorem,

$$g(\zeta) = h(\gamma) + h'(\gamma)(\zeta - \gamma) + \frac{h''(\theta)(\zeta - \gamma)^2}{2} + q(\zeta), \tag{11}$$

for some $\theta \in [\gamma, \zeta]$, where $h(\gamma) \approx 1/\sqrt{d}$. It will turn out that $|h'(\gamma)| = \Theta(1/\sqrt{d})$, $|h''(\zeta)| = o(1/\sqrt{d})$. This will allow us to complete the proof using the linear approximation of $g(\zeta)$ by neglecting the second order term and $q(\zeta)$. We now compute $h', h''$, treating $\|x\| = \sqrt{d + \zeta^2}$ as a constant for exposition (the proof works without this approximation or the reader may think of $\gamma$ as $o(\sqrt{d})$). Using $\tau_i'(\zeta) \approx \frac{\gamma}{\rho_1 \|x\|}, \tau_d'(\zeta) \approx \frac{-\gamma}{\rho_2 \|x\|}$,

$$h'(\zeta) \approx a\rho_1 \left[ \sum_{i=0}^{d-1} \binom{d-1}{i} \kappa'(\tau_i(\zeta)) \frac{\gamma}{\rho_1 \|x\|} + \frac{2^{d-1}\beta_d}{\kappa(1)} \kappa'(\tau_d(\zeta)) \frac{\gamma}{\rho_2 \|x\|} \right] + \frac{\kappa'(\tau_d(\zeta))}{\rho_2 \kappa(1)} \frac{\gamma}{\rho_2 \|x\|}$$

$$h''(\zeta) \approx a\rho_1 \left[ \sum_{i=0}^{d-1} \binom{d-1}{i} \kappa''(\tau_i(\zeta)) \frac{\gamma^2}{\rho_1^2 \|x\|^2} - \frac{2^{d-1}\beta_d}{\kappa(1)} \kappa''(\tau_d(\zeta)) \frac{\gamma^2}{\rho_2^2 \|x\|^2} \right] - \frac{\kappa''(\tau_d(\zeta))}{\rho_2 \kappa(1)} \frac{\gamma^2}{\rho_2^2 \|x\|^2}.$$

Plugging $\|x\| \approx \rho_1 \approx \sqrt{d}$ and substituting $a$ from eqn. (9),

$$h'(\zeta) = \frac{(1 + \beta_d^2/\xi)\kappa'(\tau_d(\zeta))\gamma}{\rho_2^2 \kappa(1)\sqrt{d}} + o\left(\frac{1}{\sqrt{d}}\right) \qquad \text{and} \qquad h''(\zeta) = O\left(\frac{1}{d}\right),$$

which substituted in eqn. (11) with $\tau_d(\zeta) \approx 0, \beta_d \approx \kappa(0), \kappa'(\tau_d(\zeta)) \approx \kappa'(0)$ gives

$$g(\zeta) = \frac{1}{\sqrt{d}} \left( 1 + \frac{(1 + \kappa^2(0)/\xi)\kappa'(0)\gamma}{\kappa(1)\rho_2^2}(\zeta - \gamma) \right) + o\left(\frac{1}{\sqrt{d}}\right),$$

Hence, $g(\zeta) > 0$ whenever the coefficient of $1/\sqrt{d}$ above is bounded above zero, and a similar condition holds for $g(\zeta) < 0$. Using the estimates of $\xi$ from eqn. (9) and $\kappa'(0) = 1, \kappa(0) = 1/\pi, \kappa(1) = 2, \rho_2^2 = 1 + \gamma^2$ in the above gives that $g(\zeta) > 0$ for $\zeta > -0.68\gamma - 1.68/\gamma$ and $g(\zeta) < 0$ for $\zeta < -0.905\gamma - 1.905/\gamma$.

**Predicted label for point $(\zeta, x_{2:d+1}^{(t)})$ where $x^{(t)} \in D$ has negative label**

Following the same plan, write our point (denoted by $x$) as $(\zeta, 0, \ldots, 0, 1)$.

*Explicit form for $f$.* Begin by finding

$$\tau_i = \left( \frac{1 + \gamma\zeta}{\rho_1 \|x\|} \right) \qquad \text{and} \qquad \tau_d = \left( \frac{1 - \gamma\zeta}{\rho_2 \|x\|} \right).$$

eqn. (10) now gives

$$g(\zeta) = 2^{d-1} a\rho_1 \left[ \kappa(\tau_0(\zeta)) - \frac{\beta_d \kappa(\tau_d(\zeta))}{\kappa(1)} \right] - \frac{\kappa(\tau_d(\zeta))}{\rho_2 \kappa(1)}.$$

Expanding $\kappa(\tau_0(\zeta))$ using Taylor series around $\zeta = -1/\gamma$,

$$\kappa(\tau_0(\zeta)) = \kappa(0) + \kappa'(\tau_0(\theta))\tau_0'(\theta)(\zeta + \frac{1}{\gamma}),$$

for some $\theta \in [-1, 1]$. For large $d$, $\tau_0(\theta) \approx 0$ and $\tau_0'(\theta) = O(1/\sqrt{d})$. Hence we have

$$g(\zeta) = \frac{\rho_2 \kappa(1) + \rho_1 \beta_d}{\xi \rho_1 \rho_2} \left[ \kappa(0) + O\left(\frac{1}{\sqrt{d}}\right) - \frac{\beta_d \kappa(\tau_d(\zeta))}{\kappa(1)} \right] - \frac{\kappa(\tau_d(\zeta))}{\rho_2 \kappa(1)}$$

$$= \frac{1}{\rho_2} \left( \frac{\kappa^2(0)}{\xi} - \left( \frac{\kappa^2(0)}{\xi\kappa(1)} + \frac{1}{\kappa(1)} \right) \kappa(\tau_d(\zeta)) \right) + o(1).$$

As before $g(\zeta) > 0$ whenever the coefficient of $1/\rho_2$ above is bounded above zero which happens for $\zeta \geq 0.73$ (for $\gamma \geq 3$). Similarly, $g(\zeta) < 0$ for $\zeta \leq 0$. $\qquad \square$

# B  EXPERIMENTS

In this section, we provide experimental details, including hyperparameter tuning setup and some additional experiments.

## B.1  DETAILS ON THE EXPERIMENTAL SETTING

We will first describe the four datasets that have been used in this work.

1. **Imagenette** (FastAI, 2021): This is a subset of 10 classes of Imagenet, that are comparatively easier to classify.
2. **b-Imagenette**: This is a binarized version of Imagenette, where only a subset of two classes (tench and English springer) is used.
3. **Waterbirds-Landbirds** (Sagawa et al., 2020a): This is a majority-minority group dataset, consisting of waterbirds on water and land background, as well as landbirds on land and water background. This dataset serves as a baseline for checking the dependence of model on the spurious background feature when predicting the bird class, as most of the training examples have waterbirds on water and landbirds on land background.
4. **MNIST-CIFAR** (Shah et al., 2020): This is a collage dataset, created by concatenating MNIST and CIFAR images along an axis. This is a synthetic dataset for evaluating the simplicity bias of a trained model.

**Setup**  Throughout the paper, we work with the pretrained representations of the above datasets, obtained by using an Imagenet pretrained Resnet 50. We finetune a 1-hidden layer FCN (hidden dimension - 100) on top of these representations (keeping the backbone fixed) using SGD with a momentum of 0.9. Every model is trained for 20000 steps (large enough for convergence) with a warmup and cosine decay learning rate scheduler. For each of the runs, we tune the batch size, learning rate and weight decay using validation accuracy. Below are the hyperparameter tuning details:

- Batch size $\in \{128, 256\}$
- Learning rate:
  - Rich regime: $\in \{0.5, 1.0\}$ (as learning rate in rich regime needs to scale up with the hidden dimension)
  - Lazy regime: $\in \{0.01, 0.05\}$
- Weight decay: $\in \{0, 1e^{-4}\}$

The final numbers reported are averaged across 3 independent runs with the selected hyperparameters.

**Evaluation**  For Imagenette, b-Imagenette and MNIST-CIFAR, we report the standard test accuracy in all the experiments. For waterbirds, we report train-adjusted test accuracy, as reported in Sagawa et al. (2020a). Precisely, accuracy for each group present in the test data is individually calculated and then weighed by the proportion of the corresponding group in the train dataset.

## B.2  ADDITIONAL EXPERIMENTAL RESULTS

In this section, we present a few additional experimental results.

**Accuracy of $f_{\text{proj}}$**  In Table 4, we show the test accuracy of $f_{\text{proj}}$. As can be seen, even after projecting out the principal components used by $f$, $f_{\text{proj}}$ attains significantly high accuracy. Note that, in these experiments, model 1 was kept fixed and the accuracy of $f_{\text{proj}}$ is averaged across 3 runs.

**Results on Imagenet**  We trained a 1-hidden layer FCN (with 2000 hidden neurons) on Imagenet dataset, using rich regime initialization, with learning rate selected from the set - $\{5, 10\}$ (as learning rate in rich regime needs to scale up with hidden dimension). The evolution of effective rank of the

Table 4: Trained accuracy of $f_{\text{proj}}$ in rich regime

| Dataset | Acc($f$) | Acc($f_{\text{proj}}$) |
|---------|----------|------------------------|
| b-Imagenette | 93.35 | $91.35 \pm 0.32$ |
| Imagenette | 79.67 | $71.93 \pm 0.12$ |
| Waterbirds | 90.29 | $89.92 \pm 0.08$ |
| MNIST-CIFAR | 99.69 | $98.95 \pm 0.02$ |
| Imagenet | 72.02 | $69.63 \pm 0.08$ |

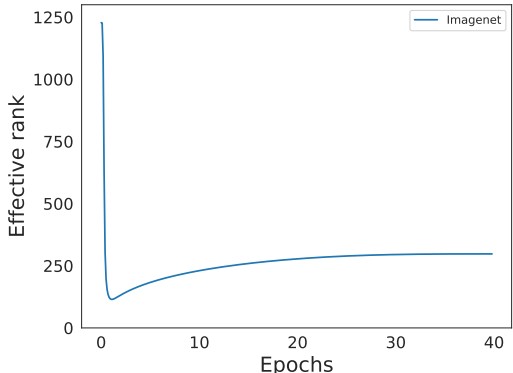

Figure 6: Evolution of effective rank of first layer weight matrix (dimension - $2048 \times 2000$) for Imagenet dataset in rich regime.

first layer weight matrix is shown in Figure 6. As can be seen, the weight matrix becomes sufficiently low rank as the training progresses. Also, $P$ and $P_\perp$ randomized accuracy (RA) and Logit change (LC) are shown in Table 5. As can be seen, the model's prediction is almost determined by the projection along the top 150 singular vectors of the weight matrix.

We also train a model2 on representations obtained by projecting out the top 150 singular vectors of the weight matrix of model 1. In Table 6, we show the mistake diversity (Mist-Div) and class-conditioned logit correlation (CC-LogitCorr) between model 1 and model 2. As can be seen, the projected out model has higher diversity and lower correlation as compared to an independently trained model. In Table 4, we also show that the second model achieves comparable accuracy to model 1.

**Singular value decay** . In Figure 7, we provide the singular value decay of the weight matrix for the first model trained in rich regime. As can be seen, the top few singular values capture most of the Frobenius norm of the matrix.

**MNIST-CIFAR** In Figure 8, we show that an ensemble of $f$ and $f_{\text{proj}}$ has better gaussian robustness than an ensemble of $f$ and $f_{\text{ind}}$ on MNIST-CIFAR dataset.

**Quantitative measurement of non-linearity of decision boundary** In this section, we report a quantitative measure of non-linearity of the decision boundary along the top two singular vectors for $f$ and $f_{\text{proj}}$. Basically, we fit a linear classifier to the decision boundary and report its accuracy. As shown in Table 7, the test accuracy obtained by the linear classifier for $f_{\text{proj}}$ is less than $f$.

**Variation of LD-SB with depth** In Figure 9 and 10, we show the evolution of effective rank of weight matrices for depth-2 and 3 ReLU networks. As can be seen, the rank still decreases with training, however the effect is less pronounced for the initial layers. Note that the initialization used in these runs was the feature learning initialization as proposed in Yang & Hu (2021).

Table 5: Demonstration of LD-SB in the rich regime: This table presents $P_\perp$ and $P$ randomized accuracies (RA) as well as logit changes (LC) on the Imagenet dataset. These results confirm that projection of input $x$ onto the subspace spanned by $P$ essentially determines the model's prediction on $x$. ↑ (resp. ↓) indicates that LD-SB implies a large (resp. small) value.

| Dataset | rank $(P)$ | Acc$(f)$ | $P_\perp$-RA (↑) | $P$-RA (↓) | $P_\perp$-LC (↓) | $P$-LC (↑) |
|---|---|---|---|---|---|---|
| Imagenet | 150 | 72.15±0.09 | 68.35±0.04 | 0.25 ± 0.0 | 1901.27 ± 0.2 | 199958 ± 44.51 |

Table 6: Mistake diversity and class conditioned logit correlation of models trained independently (Mist-Div $(f, f_{\text{ind}})$ and CC-LogitCorr $(f, f_{\text{ind}})$ resp.) vs trained sequentially after projecting out important features of the first model (Mist-Div $(f, f_{\text{proj}})$ and CC-LogitCorr $(f, f_{\text{proj}})$ resp.). The results demonstrate that $f$ and $f_{\text{proj}}$ are more diverse compared to $f$ and $f_{\text{ind}}$.

| Dataset | Mist-Div $(f, f_{\text{ind}})$ (↑) | Mist-Div $(f, f_{\text{proj}})$ (↑) | CC-LogitCorr $(f, f_{\text{ind}})$ (↓) | CC-LogitCorr $(f, f_{\text{proj}})$ (↓) |
|---|---|---|---|---|
| Imagenet | $6.97 \pm 0.06$ | $12.31 \pm 0.16$ | $99.5 \pm 0.0$ | $92.52 \pm 0.01$ |

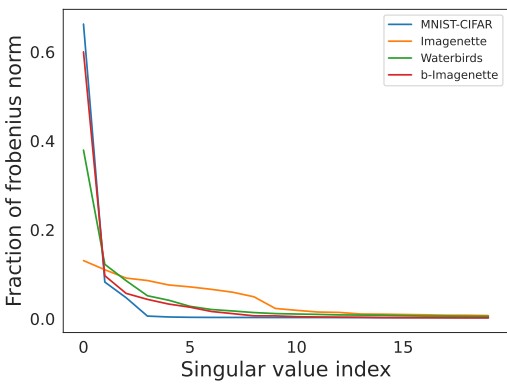

Figure 7: Fraction of Frobenius norm captured by the top $i^{th}$ singular value i.e., $\sigma_i^2 / \sum_{j=1}^{d} \sigma_j^2$ vs $i$ of the first layer weight matrix trained in rich regime for various datasets.

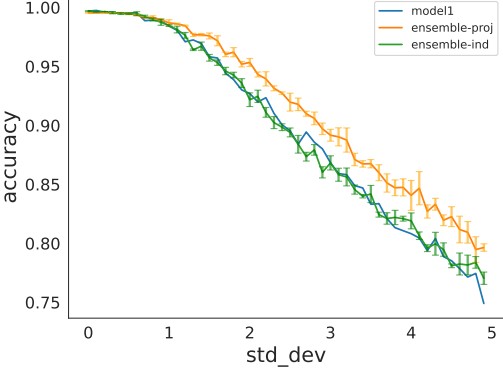

Figure 8: Variation of test accuracy with the standard deviation of Gaussian noise added to the pretrained representations of MNIST-CIFAR dataset. Model 1 is kept fixed, and values for both the ensembles are averaged across 3 runs.

Table 7: Quantitative measurement of non-linearity of decision boundary – accuracy of fitted linear classifier to the decision boundary

| Dataset | Linear-Classifier-Acc($f$) | Linear-Classifier-Acc($f_{\text{proj}}$) |
|---|---|---|
| b-Imagenette | 96.12 | $95.28 \pm 0.2$ |
| Waterbirds | 97.28 | $93.24 \pm 0.24$ |

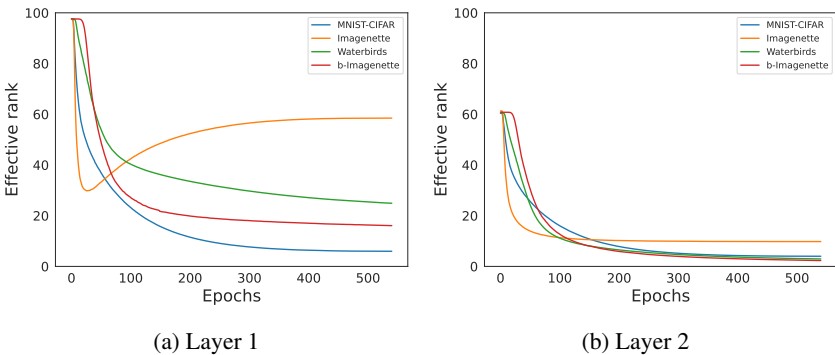

(a) Layer 1          (b) Layer 2

Figure 9: Evolution of effective rank of the weight matrices for a depth-2 ReLU network on Resnet-50 pretrained representations of the dataset

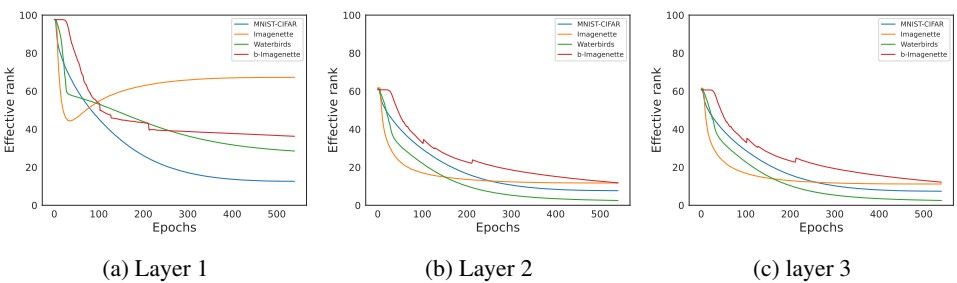

(a) Layer 1      (b) Layer 2      (c) layer 3

Figure 10: Evolution of effective rank of the weight matrices for a depth-3 ReLU network on Resnet-50 pretrained representations of the dataset

## C  EXTENDED RELATED WORKS

In this section, we provide an extensive literature survey of various topics that the paper is based on.

**Low rank Simplicity Bias in Linear Networks**   Multiple works have established low rank simplicity bias for gradient descent on linear networks, both for squared loss as well as cross-entropy loss. For squared loss, Gunasekar et al. (2017) conjectured that the network is biased towards finding minimum nuclear norm solutions for two-layer linear networks. Arora et al. (2019) refuted the conjecture and instead argued that the network is biased towards finding low rank solutions. Razin & Cohen (2020) provided empirical support to the low rank conjecture, by providing synthetic examples where the network drives nuclear norm to infinity, but minimizes the rank of the effective linear mapping. Li et al. (2021) established that for small enough initialization, gradient flow on linear networks follows greedy low-rank learning trajectory. For binary classification on linearly separable data, Ji & Telgarsky (2019) showed that the weight matrices of a linear network eventually become rank-1 as training progresses.

**Low rank Simplicity Bias in Non-Linear Networks**   For non-linear networks, the work related to low-rank simplicity bias is rather sparse. Two of the most notable works are Huh et al. (2021) and Galanti & Poggio (2022). Huh et al. (2021) empirically established that the rank of the embeddings learnt by a neural network with ReLU activations goes down as training progresses. Galanti & Poggio (2022) provided an intuition behind the relation between the rank of the weight matrices and various hyperparameter such as batch size, weight decay etc. In contrast to these works, for 1 layer nets, we theoretically and empirically establish that the network depends on an extremely low dimensional projection of the input, and this bias can be utilized to develop a robust classifier.

**Relation to OOD**   Many recent works in OOD detection (Cook et al., 2020; Zaeemzadeh et al., 2021) explicitly create low-rank embeddings so that it is easier to discriminate them for an OOD point. Other works also implicitly rely on the low-rank nature of the embeddings. Ndiour et al. (2020) use PCA on the learnt features, and only model the likelihood along the small subspace spanned by the top few directions. Wang et al. (2022) utilise the low rank nature of the embeddings to estimate the perpendicular projection of a given data point to this low rank subspace and combine it with logit information to detect OOD datapoints. While there have been works implicitly utilizing the low rank property of embeddings, we note that our paper (i) demonstrates low rank property of the *weights*, rather than that of embeddings, and (ii) shows that it is a consequence of SB.

**Other Simplicity Bias**   There have been many works exploring the nature of simplicity bias in neural networks, both empirically and theoretically. Kalimeris et al. (2019) empirically demonstrated that SGD on neural networks gradually learns functions of increasing complexity. Rahaman et al. (2018) empirically demonstrated that neural networks tend to learn lower frequency functions first. Ronen et al. (2019) theoretically established that in NTK regime, the convergence rate depends on the eigenvalues of the kernel spectrum. Hacohen et al. (2020) showed that neural networks always learn train and test examples almost in the same order, irrespective of the architecture. Pezeshki et al. (2021) proposes that *gradient starvation* at the beginning of training is a potential reason for SB in the lazy/NTK regime but the conditions are hard to interpret. In contrast, our results are shown for any dataset in the IFM model in the *rich* regime of training. Lyu et al. (2021) consider antisymmetric datasets and show that single hidden layer input homogeneous networks (i.e., without *bias* parameters) converge to linear classifiers. However, such networks have strictly weaker expressive power compared to those with bias parameters. Hacohen & Weinshall (2022) showed that for deep linear networks, in NTK regime, they learn the higher principal components of the input data first. Most of the previous works used simplicity bias as a reason behind better generalization of neural nets. However, Shah et al. (2020) showed that extreme simplicity bias could also lead to worse OOD performance.

**Learning diverse classifiers**: There have been several works that attempt to learn diverse classifiers. Most works try to learn such models by ensuring that the input gradients of these models do not align (Ross & Doshi-Velez, 2018; Teney et al., 2022). Xu et al. (2022) proposes a way to learn diverse/orthogonal classifiers under the assumption that a complete classifier, that uses all features is available, and demonstrates its utility for various downstream tasks such as style transfer. Lee et al. (2022) learns diverse classifiers by enforcing diversity on unlabeled target data.

**Spurious correlations**: There has been a large body of work which identifies the reasons for spurious correlations in NNs (Sagawa et al., 2020b) as well as proposing algorithmic fixes in different settings (Liu et al., 2021; Chen et al., 2020).

**Implicit bias of gradient descent**: There is also a large body of work understanding the implicit bias of gradient descent dynamics. Most of these works are for standard linear (Ji & Telgarsky, 2019) or deep linear networks (Soudry et al., 2018; Gunasekar et al., 2018). For nonlinear neural networks, one of the well-known results is for the case of 1-hidden layer neural networks with homogeneous activation functions (Chizat & Bach, 2020), which we crucially use in our proofs.

