# OpenReview forum: "Simplicity bias in $1$-hidden layer neural networks"
_ICLR.cc/2023/Conference — Submitted to ICLR 2023_

### Official Review · Reviewer_iV59 · 2022-10-24

**Confidence:** 3
**Correctness:** 3
**Technical Novelty And Significance:** 3
**Empirical Novelty And Significance:** 2
**Recommendation:** 5

**Clarity, Quality, Novelty And Reproducibility:**

Previous work (that is already cited in Geirhos et al) used the term shortcut learning, so I am wondering why the term has been changed to simplicity bias here. Won’t this obfuscate the literature if there are two terms describing the same phenomena?

Is there any proof that the ensemble avoids the SB?

At the moment, I find the empirical validation, especially in the part of the ensemble method, quite weak. For instance, I am wondering why in Table 1 only up to 10 classes are used. Isn’t it possible that by needing to learn more classes, e.g. in Imagenet, the network avoids such shortcut learning?

Minor: The text requires some proofreading since there inconsistencies, e.g. “Hu et al observe”, but “Lyu et al considers”; I think all references need to be corrected into the former format.


**Strength And Weaknesses:**

 * [+] Understanding of neural networks and their (learning) properties is an important task.

 * [+] The definition of simplicity bias in Definition 1.1 is clear to me.

 * [-] The results are currently limited, e.g. only one-hidden layer network. Despite the authors explicitly identifying this, I am still not convinced that one-hidden layer nets are a practical consideration.

 * [-] The core theorem 4.1 is only applicable for an infinite width network.

 * [-] The paper mentions that we do not have a clear understanding of the dynamics of deeper networks (page 2), however this is not true. The papers of “Training Integrable Parameterizations of Deep Neural Networks in the Infinite-Width Limit” or “Gradient Descent Finds Global Minima of Deep Neural Networks” already study the dynamics of deep networks (or their proof applies for deep nets). I would recommend revising this paragraph to avoid any confusion from the readers.


**Summary Of The Paper:**

The paper explores the simplicity bias (SB) for 1-hidden layer neural networks (NNs). The paper formulates SB as a concept (the NN learns a simpler feature than those available and informative for classification) and then prove that this might happen in 1-hidden layer NNs. Lastly, the paper proposes a heuristic way to train an ensemble of models to avoid SB.

**Summary Of The Review:**

At the moment, there are certain unclear things to me, e.g. whether the simplicity bias is the same as shortcut learning, or why the results cannot be extended to deep networks or the finite width regime. In addition, I find the experiments are rather weak. Having said that, it is possible that this paper has value for the community if the paper is strengthened during the rebuttal period.

---

> ### Author Response · Authors · 2022-11-11
> **Response to Reviewer iV59**
>
> We thank the reviewer for their time and effort in reviewing our paper, and for the valuable feedback. Our responses below:
>
> 1. **Utility of one-hidden layer networks**: As pretraining on large and diverse corpuses and then fine-tuning on the target datasets is one of the dominant paradigms in machine learning, we believe that our results on one-hidden layer networks are quite relevant. Particularly in the context of OOD robustness, there have been several recent works which suggest that freezing the feature extractor backbone and training a shallow (or even linear) classifier might be much better compared to full finetuning, reinforcing our position that fine-tuning a simple head on top of pre-trained models is a useful paradigm in practical machine learning.
>
> &nbsp;&nbsp;&nbsp;&nbsp;&nbsp;&nbsp;&nbsp;[1] Fine-Tuning can Distort Pretrained Features and Underperform Out-of-Distribution by Ananya Kumar et al., 2022
>
> &nbsp;&nbsp;&nbsp;&nbsp;&nbsp;&nbsp;&nbsp;[2] Domain-adjusted regression or: Erm may already learn features sufficient for out-of-distribution generalization by Elan Rosenfeld et al., 2022
>
> &nbsp;&nbsp;&nbsp;&nbsp;&nbsp;&nbsp;&nbsp;[3] DAFT: Distilling Adversarially Fine-tuned Models for Better OOD Generalization by Anshul Nasery et al., 2022
>
> 2. **Extension to finite width networks and larger depth**: We agree with the reviewer that extending these ideas to finite width networks and for deeper networks is an extremely important direction. However, as noted in the paper a *characterization of asymptotic convergence point of gradient descent* is available only for inifinite width, one hidden layer networks at the present. While there are several papers, as shared by the reviewer, that show global convergence for deeper networks as well, a *precise characterisation of the global minima that the gradient descent trajectory converges to* is only known for one hidden layer networks. One approach could be to try to characterize the limit point of gradient descent of deeper models *for IFM datasets* (as opposed to any general dataset) and extend our results. This is indeed an interesting direction of future work, and would require substantial new ideas.
>
> 3. **Shortcut learning vs SB**: We use shortcut learning (SL) and SB interchangeably in the paper. To be more precise, SL is the tendency of neural networks to learn features that do not generalize well to OOD settings, as described in the survey paper [1]. On the other hand, SB is the *mathematical reason* why neural networks tend to suffer from shortcut learning [2].
>
> &nbsp;&nbsp;&nbsp;&nbsp;&nbsp;&nbsp;&nbsp;[1] Shortcut Learning in Deep Neural Networks by Robert Geirhos et al., 2020
>
> &nbsp;&nbsp;&nbsp;&nbsp;&nbsp;&nbsp;&nbsp;[2] The Pitfalls of Simplicity Bias in Neural Networks by Harshay Shah et al., 2020
>
> 4. **Whether ensemble avoids SB**: The approach we took to assess this was by checking the robustness of the ensemble. As shown in Figure 5, the Gaussian robustness of the ensemble is much better as compared to the first model. We would love to hear if the reviewer has any other suggestions.
>
> 5. **Empirical datasets have only 10 classes**: We have initiated experiments on the full imagenet dataset and will update our response once we have those results.

---

> > ### Comment · Reviewer_iV59 · 2022-11-12
> > **Further clarifications**
> >
> > I am thankful to the authors for responding to the original review. Upon studying their answers, I still have few questions:
> >
> > * Firstly, could the authors clarify exactly what do they mean by "precise characterisation of deeper networks"? What are the parts in the analysis of the dynamics are required and do not hold for deeper networks?
> >
> > * Additionally, why is the linearly separable assumption realistic, especially given the low-rank assumptions in the paper?

---

> > > ### Author Response · Authors · 2022-11-14
> > > **Response to Reviewer iV59**
> > >
> > > We thank the reviewer for going through the response. Regarding the comments:
> > >
> > > 1. **Precise characterisation of deep nets**: [1] showed that, for one hidden layer FCN, when trained using gradient descent on cross entropy loss, it converges in direction to the F1-max margin classifier of the dataset. We use this characterisation in the proof, to argue that the network will eventually utilize just the linear coordinate for classification. For deep nets, although results about global convergence exist, a *characterisation* of the convergence point (there could be several global minima) of gradient descent trajectory is missing. Here, by characterization, we mean the solution to some optimization problem, like the F1-max margin classifier.
> > >
> > > &nbsp;&nbsp;&nbsp;&nbsp;&nbsp;&nbsp;&nbsp;[1] - Lenaic Chizat and Francis R. Bach. - Implicit Bias of Gradient Descent For Wide Two-layer neural networks trained with the logistic loss
> > >
> > > 2. **Linear Separability**: Since we are considering **pretrained** representations, they tend to be (almost) linearly separable – e.g., most transfer learning/self supervised learning papers report **linear evaluation** accuracy on the target task which can be as high as 75% on full ImageNet (e.g., see Table 1 in [2]).We also note that the low rank part is *not an assumption*. The main contribution of our paper is to *prove* that the low rank dependence arises as a consequence of SB.
> > >
> > > &nbsp;&nbsp;&nbsp;&nbsp;&nbsp;&nbsp;&nbsp;[2] Barlow Twins: Self-Supervised Learning via Redundancy Reduction by Zbontar et al., 2021

---

> > > > ### Author Response · Authors · 2022-11-18
> > > > **Imagenet results**
> > > >
> > > > We have updated Appendix B.2 with results on the Imagenet dataset. We used a 1-hidden layer FCN (with 2000 neurons in hidden dimension) on top of pretrained resnet-50 representations (dimension - 2048).
> > > >
> > > > 1. In Figure 6, we show the evolution of the effective rank of the first layer weight matrix. As can be seen, the weight matrix becomes sufficiently low rank as the training progresses.
> > > > 2. In Table 5, we show the results regarding $P$ and $P_{\perp}$ randomized accuracy (RA) and logit correlation (LC) for this model, where $P$ is chosen to be the space spanned by the top $150$ singular vectors of the weight matrix. As can be seen, the predictions of the model mostly depend on the projection along these top 150 singular vectors of the weight matrix .
> > > >
> > > > The training of model 2, obtained after projecting out the top 150 singular vectors of the first model’s weight matrix, is in progress. We will report the results as soon as the run finishes.

---

> > > > > ### Author Response · Authors · 2022-11-19
> > > > > **Update on Imagenet results**
> > > > >
> > > > > We also train a model 2 on representations obtained by projecting out the top 150 singular vectors of the weight matrix of model 1.
> > > > >
> > > > > 1. In Table 6, we show the mistake diversity and class-conditional logit correlation between model 1 and model 2. As can be seen, the projected out model has higher diversity and lower correlation as compared to an independently trained model.
> > > > > 2. In Table 4, we also show that the second model achieves comparable accuracy to model 1.

---

> > > > > > ### Comment · Reviewer_iV59 · 2022-11-23
> > > > > > **Response on the current state of the manuscript**
> > > > > >
> > > > > > Dear authors,
> > > > > >
> > > > > > I am thankful for providing additional experimental results on Imagenet. I notice that the explanations provided in the rebuttal for the shortcut learning and the exact reasons that the existing method is not extended to finite-width NTK or the deep net are not mentioned (even on the supplementary). Do the authors plan on including them?
> > > > > >
> > > > > > There are some other minor details that require to be fixed as well. For instance, Imagenet is not described in the experimental details in sec. B.1.
> > > > > >
> > > > > > In addition, the writing style around references is not consistent. This was already mentioned in the original review, but never fixed. For instance, "Hu et al empirically observe ", but "Xu et al proposeS" or "Lee et al learnS". All of these cases are in the same page as originally mentioned in the review.

---

> > > > > > > ### Author Response · Authors · 2022-11-24
> > > > > > > **Response to Reviewer iV59**
> > > > > > >
> > > > > > > Thanks a lot for going through the manuscript again.
> > > > > > >
> > > > > > > 1. **Shortcut learning and extension to finite width/deep nets** - We will make sure to include these points in the final revision.
> > > > > > > 2. **Description of Imagenet in Appendix B.1** - Currently, we kept imagenet along with the other additional experiments conducted during the rebuttal period in Appendix B.2. In the final revision, we will move it to Appendix B.1
> > > > > > > 3. **Writing style** - Since our focus was on addressing the concerns about results and analysis, we missed addressing the writing style related aspects. Sorry about that and thanks for reminding us again. We will make sure to address these in the final revision.

---

### Official Review · Reviewer_EewH · 2022-10-25

**Confidence:** 4
**Correctness:** 3
**Technical Novelty And Significance:** 2
**Empirical Novelty And Significance:** 3
**Recommendation:** 6

**Clarity, Quality, Novelty And Reproducibility:**

The paper is well written and the idea is smoothly presented.

The proposed LD-SB is novel and interesting.

**Strength And Weaknesses:**

Strength:

- The concept of LD-SB is clearly and rigorously defined, along with a good presentation of related background knowledge.
- Both theoretical proof (for 1-hidden layer net and IFM) and empirical justification (for four real datasets) are provided to demonstrate the LD-SB.
- The paper further proposed a potentially useful ensemble method, OrthoP, and show this method is more robust under Gaussian noise.


Weaknesses:

- In the experiments on real datasets, the current four metrics do not reflect the third point of the definition for LD-SB. Hence, the current experiments do not fully “show LD-SB on real datasets.”  As the projection operator is ready, why not train an independent model to show that a model using the remaining features in the orthogonal sub-space can achieve high accuracy?
- Despite the authors' arguments that fine-tuning on 1-hidden layer network is effective in many practical cases, I do think for real datasets, experiments could involve more sophisticated network architectures. It is interesting to see whether the LD-SB exists in more complex and practical settings.
- In subsection 4.4.3 for Lazy regime, it said that “able to find a low dimensional projection matrix P which satisfies the conclusions in Theorem 4.2. This statement seems improper since Theorem 4.2 applies to a specific dataset rather than those real datasets.
- Can you discuss why the low dimension character is also present when applying to real datasets rather than IFM? Is there any other mechanism involved?
- Missing details of experiments referred to in Fig.5. For example, how is the output of the ensemble model computed? Average over both member models?
- In Fig.4, it is claimed that the decision boundary of the second model is more non-linear. It would be better to provide a quantitative measure of the non-linearity.


**Summary Of The Paper:**

This paper elaborates on a specific simplicity bias called LD-SB. Theoretical proofs of LD-SB are provided for 1-hidden layer neural networks on the IFM distribution under different situations. Experiments on four real datasets demonstrate that LD-SB is of practical importance. The authors further propose a new robust ensemble method based on the understanding of LD-SB.

**Summary Of The Review:**

The proposed LD-SB is interesting and backed by both theoretical and empirical justifications. However, according to my understanding, the experiments on real datasets do not fully establish the LD-SB (see point 1 in weaknesses). Hence, I give a score of 5, e.g., marginally below the threshold.

------------------------20221122-----------------------

as the authors had addressed my concern about establishing LD-SB in real datasets, I decided to raise my rating from 5 to 6.

---

> ### Author Response · Authors · 2022-11-11
> **Response to Reviewer EewH**
>
> We thank the reviewer for their time and effort in reviewing our paper, and for providing valuable feedback. Regarding the comments:
>
> 1. **High accuracy of model 2**: We have added Table 4 in Appendix B.2, demonstrating the accuracy of model 2, trained after projecting out the top subspaces of model 1. As can be seen, these attain a significant accuracy as well.
>
> 2. **More complex architectures**: While prior work e.g., [1] has shown SB even in more complex architectures, the precise form of SB is indeed unknown. A natural conjecture based on LD-SB would be that each layer of a deeper network exhibits LD-SB. In Figure 8 and 9 in Appendix B.2, we conduct experiments on multi layer networks, and observe that while LD-SB goes down with depth, it is still present. While we have made progress, by identifying the nature of SB in 1-hidden layer networks, thoroughly extending this to deeper architectures is an interesting direction of future work.
>
> &nbsp;&nbsp;&nbsp;&nbsp;&nbsp;&nbsp;&nbsp;[1] Shortcut Learning in Deep Neural Networks by Robert Geirhos et al., 2020
>
> 3. **Subsection 4.4.3 and Theorem 4.2**: Thanks for pointing this out. We have updated the draft to clarify that the empirical results satisfy Definition 1.1 (instead of Theorem 4.2).
>
> 4. **LD-SB in real datasets vs in IFM**: We believe that the mechanism of LD-SB in real datasets is for the same reason it happens in IFM. For 2-homogeneous networks, $F_1$ max-margin-classifier implicitly behaves as a $L_1$ regularization on the weights (See Section 1 of [1]). Thus, it has a bias towards relying on only a few features, even in the presence of several useful features. Our proof makes this intuition rigorous.
>
> 	[1] Implicit Bias of Gradient Descent for wide two layer neural networks -
> https://francisbach.com/gradient-descent-for-wide-two-layer-neural-networks-implicit-bias/
>
> 5. **Missing detail in Figure 5**: Thanks for pointing this out. Yes, we average the two models in the logit space. We have added this detail to the revised version.
>
> 6. **Quantitative measure of non-linearity**: Thanks again for the question. One way is to fit a linear classifier to the decision boundary, and measure its accuracy. These numbers are now reported in Table 5 in Appendix B.2. Please let us know if you have any other suggestions on how to do this.

---

> > ### Author Response · Authors · 2022-11-19
> > **Further feedback**
> >
> > Dear Reviewer EewH
> >
> > We have submitted our response to the concerns raised during the review period. We would be really grateful if you can take a look at it, and provide us further feedback.
> >
> > Thanks

---

> > ### Comment · Reviewer_EewH · 2022-11-21
> > **Thanks for the reply**
> >
> > I appreciate the detailed reply and am happy to see my main concerns are well addressed. Hence, I decided to raise my rating from 5 to 6.

---

### Official Review · Reviewer_Hiqj · 2022-10-27

**Confidence:** 4
**Correctness:** 4
**Technical Novelty And Significance:** 2
**Empirical Novelty And Significance:** 2
**Recommendation:** 5

**Clarity, Quality, Novelty And Reproducibility:**

The low-rank idea is NOT new for defining simplicity bias, therefore, the paper needs to be polished to distinguish from previous works.

**Details Of Ethics Concerns:**

N.A.

**Strength And Weaknesses:**

Strengths:

S1. The paper is very well-written and I enjoyed reading it.
S2. Both theory and experiments look great.

Weaknesses:

S1. The low-rank bias is NOT new. Please refer to [R1] and [R2] as well as related works in the two papers.

[R1] The Low-Rank Simplicity Bias in Deep Networks. https://minyoungg.github.io/overparam/resources/overparam-v2.pdf
[R2] Principal Components Bias in Over-parameterized Linear Models, and its Manifestation in Deep Neural Networks. JMLR 2022

S2. A better overview of previous related work is suggested. E.g., as mentioned in Shah et al. (2020), simplicity bias is related to Out-of-Distribution (OOD) detection. In OOD detection, the low-rank bias and ensemble learning along the orthogonal projection has also been extensively studied.

[R3] Outlier detection through null space analysis of neural networks. arXiv:2007.01263, 2020.

[R4] Out-of-distribution detection with subspace techniques and probabilistic modeling of features. arXiv:2012.04250, 2020.

[R5] Out-of-distribution detection using union of 1-dimensional subspaces. CVPR 2021.

[R6] ViM: Out-Of-Distribution with Virtual-logit Matching, CVPR 2022.

**Summary Of The Paper:**

This paper rigorously defined as well as thoroughly established simplicity bias for one hidden layer neural networks, as a function of a low dimensional projection of the inputs. The authors theoretically proved that the network primarily depends on only the linearly separable subspace when the data is linearly separable, and emprically showed that the models trained on real datasets depended on a low dimensional projection of the inputs. Moreover, they proposed an ensemble approach to combine the orthogonal direction learning.

**Summary Of The Review:**

This paper rigorously defined as well as thoroughly established simplicity bias for one hidden layer neural networks, as a function of a low dimensional projection of the inputs. The low-rank idea is NOT new for defining simplicity bias, therefore, the paper needs to be polished to distinguish from previous works.

---

> ### Author Response · Authors · 2022-11-11
> **Response to Reviewer Hiqj**
>
> We thank the reviewer for their time and effort in reviewing our paper, and for providing valuable feedback and related references. Regarding the comments:
>
> 1. **Low rank bias not new and novelty**: We have cited [R1] in related works, and noted that the paper empirically demonstrates that the *embeddings* learned by neural networks have a low-rank structure, primarily on synthetic datasets. In contrast, in our paper, we **theoretically** and **empirically** establish that, for one hidden layer networks, the network is essentially a function of a low dimension projection of the input (*even in the lazy regime, when weight matrices are themselves not low-rank*). Finally, we show how to use LD-SB to train a second model that is diverse compared to the first model, and obtain a more robust ensemble. The second work [R2] primarily focuses on deep linear nets in the NTK regime, and shows that the network learns features in order of decreasing eigenvalues of the input covariance matrix. However, they do not show that the network learns only a low dimensional projection, when trained to convergence.
>
> 2. **Better overview of previous related work**: Thank you for the related references from OOD literature. We have now added a discussion about them to Appendix C. To summarize, these works implicitly assume that the embeddings learnt by a neural network have a low rank structure (some of them modify the loss function to make embeddings low rank, which is not an implicit bias anymore). In contrast, our work thoroughly demonstrates LD-SB both theoretically and empirically.

---

> > ### Author Response · Authors · 2022-11-19
> > **Further feedback**
> >
> > Dear Reviewer Hiqj
> >
> > We have submitted our response to the concerns raised during the review period. We would be really grateful if you can take a look at it, and provide us further feedback.
> >
> > Thanks

---

### Official Review · Reviewer_3SUe · 2022-11-01

**Confidence:** 4
**Correctness:** 3
**Technical Novelty And Significance:** 4
**Empirical Novelty And Significance:** 4
**Recommendation:** 8

**Clarity, Quality, Novelty And Reproducibility:**

The paper is quite clear, important and novel. However, the algorithms for computing the projection matrices are not very clear described, and there is no mention of a code release, hence reproducibility seems low.

EDIT: the reproducibility concerns have been addressed.

**Strength And Weaknesses:**

Strengths:
- The paper is well-written
- The result is interesting, important, and as far as I know, novel

Weakness:
- No discussion of why is "simplicity bias" supposedly leads to poor OOD generalization. Wouldn't a more complicated decision boundary be even less robust than a quasi-linear one?
- How are the projection matrices computed in the experimental section? There is very little discussion and no pseudo-code. Is the weight matrix used in the rich regime? How do you learn the projection matrix in the lazy regime? Are there any constraints?

**Summary Of The Paper:**

The paper analyzes the phenomenon of "simplicity bias" in neural networks. Simplicity bias is defined here as the tendency of models to learn "simple" features (low-rank linear projections of the input space) that enable the model to solve the task even when different non-linear projections could also be used to solve the task with high accuracy. It is argued in the paper that this can result in "shortcut learning" where the model can learn simple features that correspond to "spurious correlations" and are not stable w.r.t. distribution shift.

The paper focuses on 1-hidden layer ReLU binary classifiers.

A theoretical analysis is carried out in the infinite width limit, with two different initialization regimes: a "lazy" regime that corresponds to the NTK framework, and a "rich" regime with enables training of the hidden layer and is therefore more representative of practical training regimes. Under the assumption that the data comes from a distribution where the input features are independent conditional on the output, they show that the model will project the input in a rank-1 linear manifold if it suffices to solve the problem, even when other non-linear solutions exist.

An empirical analysis is done on classification heads on top of frozen pre-trained image classification models for various image classification tasks, showing results consistent with the hypothesis.
The paper also briefly discusses how to use this simplicity bias property to train an ensemble of diverse models by sequentially projecting the data on the orthogonal space of the linear projections that the model tend to learn.

**Summary Of The Review:**

Good paper, I would give a higher score if my concerns about reproducibility were addressed.

EDIT: I am increasing my score.

---

> ### Author Response · Authors · 2022-11-11
> **Response to Reviewer 3SUe**
>
> We thank the reviewer for their time and effort in reviewing our paper, and for providing valuable feedback. Regarding the comments:
>
> 1. **OOD generalization and Simplicity Bias**: Several papers have established that simplicity bias/shortcut learning is a major reason for the poor OOD robustness of neural networks e.g., see Section 5 of the survey paper [1], and [2]. Intuitively, the reason is that, a neural network that uses a few spurious features for classification such as metal tokens in the chest for predicting pneumonia is bound to have poor OOD performance [1]. From a technical point of view, a classifier with a larger margin i.e., distance to the decision boundary can be more robust than one with a smaller margin even if it has a more complicated decision boundary.
>
> &nbsp;&nbsp;&nbsp;&nbsp;&nbsp;&nbsp;&nbsp;[1] Shortcut Learning in Deep Neural Networks by Robert Geirhos et al., 2020
>
> &nbsp;&nbsp;&nbsp;&nbsp;&nbsp;&nbsp;&nbsp;[2] The Pitfalls of Simplicity Bias in Neural Networks by Harshay Shah et al., 2020
>
> 2. **Projection Matrices**: As mentioned in the paper
> - in section 4.4.2, for the rich regime, the projection matrices are obtained using the top singular vectors of the first layer weight matrix.
> - in section 4.4.3, for the lazy regime, the projection matrices are obtained via an optimization problem over the projection matrix P given by
>
> &nbsp;&nbsp;&nbsp;&nbsp;&nbsp;&nbsp;&nbsp; $\min_{P} \frac{1}{n} \sum_{i=1}^n \left(\mathcal{L}(f(Px_i),y_i) + \lambda \mathcal{L}(f(P^{\perp}x_i), \mathcal{U}[L])\right)$
>
> &nbsp;&nbsp;&nbsp;&nbsp;&nbsp;&nbsp;&nbsp; where $\mathcal{U}[L]$ represents a uniform distribution over all the $L$ labels, $(x_1, y_1),\cdots,(x_n, y_n)$ are training examples and $\mathcal{L}(\cdot,\cdot)$ is the cross entropy loss.
>
> 3. **Code Release**: We have now submitted the code through the openreview portal and the reviewers should be able to access it. We also intend to open source the code used for all of our experiments soon.

---

> > ### Author Response · Authors · 2022-11-19
> > **Further feedback**
> >
> > Dear Reviewer 3SUe
> >
> > We have submitted our response to the concerns raised during the review period. We would be really grateful if you can take a look at it, and provide us further feedback.
> >
> > Thanks

---

> > > ### Comment · Reviewer_3SUe · 2022-12-05
> > > **Good**
> > >
> > > Thanks for your response. This addresses my concerns, I've increased my score.

---

> > > > ### Author Response · Authors · 2022-12-06
> > > > **Thank you**
> > > >
> > > > Thanks for the feedback, discussion and increasing the score.

---

### Decision · Program_Chairs · 2023-01-20

**Decision:**

Reject

**Justification For Why Not Higher Score:**

There are a variety of misgivings across reviewers and based on my own reading.

**Justification For Why Not Lower Score:**

N/A

**Metareview: Summary, Strengths And Weaknesses:**

This paper studies simplicity bias of shallow networks in two settings: those where it can be assumed that logistic loss minimization converges to a globally maximum margin solution, and RKHS solutions.  While this is an interesting paper on an important topic, overall I  agree with the misgivings of a few of the reviewers and feel this work could use a bit more time; I urge the authors to focus on reviewer suggestions and prepare a careful revision for a future venue.

Technical note: Theorem 3.1, cited from Chizat&Bach, is incorrectly stated; it requires a few more technical conditions, moreover important ones as they are exceptionally hard to prove/verify.